# Intraspecific diploidization of a halophyte root fungus drives heterosis

Zhongfeng Li[1,2,14], Zhiyong Zhu[1,2,3,14], Kun Qian[4,5,14], Boping Tang[6,14], Baocai Han[7], Zhenhui Zhong [8], Tao Fu[9], Peng Zhou [10] ✉, Eva H. Stukenbrock [11,12], Francis M. Martin [2,13] ✉ & Zhilin Yuan [1,2] ✉

How organisms respond to environmental stress is a key topic in evolutionary biology. This study focused on the genomic evolution of *Laburnicola rhizohalophila*, a dark-septate endophytic fungus from roots of a halophyte. Chromosome-level assemblies were generated from five representative isolates from structured subpopulations. The data revealed significant genomic plasticity resulting from chromosomal polymorphisms created by fusion and fission events, known as dysploidy. Analyses of genomic features, phylogenomics, and macrosynteny have provided clear evidence for the origin of intraspecific diploid-like hybrids. Notably, one diploid phenotype stood out as an outlier and exhibited a conditional fitness advantage when exposed to a range of abiotic stresses compared with its parents. By comparing the gene expression patterns in each hybrid parent triad under the four growth conditions, the mechanisms underlying growth vigor were corroborated through an analysis of transgressively upregulated genes enriched in membrane glycerolipid biosynthesis and transmembrane transporter activity. In vitro assays suggested increased membrane integrity and lipid accumulation, as well as decreased malondialdehyde production under optimal salt conditions (0.3 M NaCl) in the hybrid. These attributes have been implicated in salinity tolerance. This study supports the notion that hybridization-induced genome doubling leads to the emergence of phenotypic innovations in an extremophilic endophyte.

Plant roots harbor a diverse array of fungal species spanning various lineages, including Mucoromycotina, Glomeromycotina, Basidiomycota, and Ascomycota[1]. These symbiotic organisms play a critical role in the root mycobiome and provide tangible benefits to plants, such as improving nutrient uptake and enhancing resistance to biotic and abiotic stressors. Among these fungal groups, dark septate endophytes (DSEs) are a well-known group of root-associated fungi that are highly adaptable and thrive in extreme environments[2]. DSEs are classified as class 4 endophytes because of the presence of melanized hyphae and the formation of microsclerotium-like structures in the cortical roots[3].

The Joint Genome Institute (JGI) 1000 fungal genome projects have provided some insights into the genomic organization and symbiotic gene "toolkit" of mutualistic fungi[4]. These projects have expanded our understanding of the genomic characteristics of DSEs. Unlike the convergent evolution observed in biotrophic mycorrhizal fungi[4,5], different DSEs have evolved along distinct evolutionary trajectories[6–8]. Previous studies have focused on the taxonomy, community structure, and interactions of DSEs with host plants[8]. An essential yet underexplored aspect of DSEs is their ability to thrive under challenging environmental conditions, and how genome features, dynamics, and evolution underlie their adaptation to abiotic stresses. This study investigated a recently

discovered DSE, *Laburnicola rhizohalophila* (Ascomycota)[9], which is specifically found in salt marsh environments.

Previous studies have revealed substantial genetic diversity within populations of *L. rhizohalophila* that are capable of surviving in saline environments. Our findings revealed the presence of a genomic island that underwent positive selection, thereby promoting adaptability to salinity[10]. However, further investigation is necessary to elucidate the evolutionary processes that drive this adaptation. By constructing a chromosome-level assembly, we discovered that diploid hybrids of *L. rhizohalophila* resulted from the fusion of genetically distinct haploid lineages, a process similar to intraspecific hybridization. In fungi, hybridization-mediated genome doubling can contribute to functional and phenotypic diversification, and facilitate evolution and adaptation[11,12]. In our study, we found that a naturally occurring recombinant hybrid genotype exhibited enhanced growth under various abiotic stresses. This observation motivated us to investigate how the ploidy change in the DSE fungus improves its fitness by comparing the gene expression patterns in each hybrid-parent triad under different growth conditions. Overall, our findings provide new and valuable insights into the phenomenon of diploidization in fungal species and their role in evolutionary adaptation to saline environments.

## Results

### Population genomic analysis reveals diverged populations and sign of hybridization

To assess the population structure of *L. rhizohalophila*, we analyzed a dataset comprising 47 genome sequences, including 18 newly isolated isolates and previously sequenced genomes. After single nucleotide polymorphism (SNP) calling, a subset of 804,906 filtered SNPs was obtained. The neighbor-joining (NJ) tree revealed the presence of two distinct genetic clades, designated as Groups 4 and 5, in addition to the three previously recognized clades[10] (Fig. 1C). Splitstree analysis demonstrated substantial reticulation, indicating a high level of recombination within the *L. rhizohalophila* population, which contributed to its wide genetic diversity. This pattern suggests that the five clades are likely to exhibit panmixia (Fig. 1D).

We observed an increased number of heterozygous SNPs in Groups 4 and 5. The heterozygosity ratios for these groups were $3.23 \pm 1.62\%$ (mean $\pm$ SD) and $1.82 \pm 0.65\%$, respectively. A two-sided *t*-test indicated a significant difference in the heterozygosity ratio between the two groups ($t = 29.973$, $p < 2.2\text{E}{-}16$), with Group 4 exhibiting a significantly higher heterozygosity ratio than Group 5 (Fig. 1E). Additionally, isolates from these two groups often displayed irregular sectoring of the mycelial colonies (Fig. 1F), indicative of heterokaryon formation or haploidization of heterozygous diploids[13,14].

Furthermore, during the preliminary genome assembly from Illumina reads, we noticed that the genome size of isolates from Groups 4 and 5 (~120 Mb) was nearly twice that of the isolates from other groups. This led us to further investigate their genome sizes, which strongly suggested the formation of diploid hybrids (see details below). In addition, we generated 17-mer depth–frequency distribution curves (Fig. 2A). The profiles showed two peaks, with a smaller peak having twice the coverage of the taller peak, indicating the presence of heterozygous loci. Similarly, the alternative allele frequency profiles of JP19 and JP8 both showed a value of 0.5, providing additional evidence of their potential status as heterokaryons or diploid hybrids (Fig. 2B).

### Several lines of evidence support diploid-like genome structures

By employing flow cytometry, we determined the genomic sizes of JP19 and JP8 to be $127.3 \pm 1.5$ Mb (means $\pm$ SD) and $116.3 \pm 4.5$ Mb, respectively (Fig. 2C). In contrast, the genomic sizes for JP11, R22, and JP44 were estimated to be $67.3 \pm 0.4$ Mb, $64.4 \pm 1.2$ Mb, and $60.4 \pm 0.4$ Mb, respectively. These findings suggest that the genome sizes of JP19 and JP8 nearly doubled, and any deviation between the

final assembly size and genome size measured by flow cytometry may be attributed to collapsed repeats during the assembly process[15]. We confirmed that all the isolates had mononucleate hyphal cells, ruling out the possibility of heterokaryotic hyphae (Fig. 2D). We then measured and compared the sizes of the hyphal cells and nuclei among the five isolates. Significant differences were observed in terms of nuclear size ($\chi^2 = 558.36$, $p < 2.2\text{E}{-}16$), protoplast size ($\chi^2 = 86.84$, $p < 2.2\text{E}{-}16$), length ($F = 57.38$, $p = 1.76\text{E}{-}13$), and hyphal cell width ($\chi^2 = 104.48$, $p < 2.2\text{E}{-}16$) between two diploid and three haploid isolates (Fig. 2E).

To further validate the changes in ploidy and develop mechanistic hypotheses regarding the evolutionary trajectory and adaptation of *L. rhizohalophila*, we performed high-quality genome sequencing using a combination of PacBio HiFi and high-throughput chromosomal conformation capture (Hi-C). Improved genome quality of JP44 was observed compared to that of the previous PacBio assembly, resulting in a reduction in the number of scaffolds. The final assemblies had an average genome size of approximately 65 Mb for JP44, JP11, and R22 and approximately 130 Mb for JP19 and JP8 (Table 1). The assembled genomes exhibited high contiguity with a mean $N_{50}$ value of 3.07 Mb. Using Hi-C data, intrachromosomal interaction signals were used to categorize and order the assembled contigs into chromosome-scale scaffolds. Results revealed that the haploid isolates had 21-23 chromosomes (Fig. S1), whereas JP19 and JP8 had nearly double chromosome numbers (Fig. 3A, B). Most assembled chromosomes contained one or both putative telomeric ends that were enriched in sequence repeats (Table 1). BUSCO assessment and RNA-Seq were used to verify the completeness and accuracy of the assembled genomes. More than 90% of the predicted gene models were supported by the evidence of transcript and/or protein homology. Notably, JP19 and JP8 contained approximately twice the number of protein-coding genes identified in haploids (Fig. 3C). Along with this, we detected a high proportion of duplicated BUSCO genes (95.9% and 96.6% in JP19 and JP8, respectively) (Table 1) and found that 10,731 genes in JP19 and 10,662 genes in JP8 had two copies in their genomes. Overall, our findings confirm that JP19 and JP8 have larger genome sizes and are diploid rather than dikaryotic.

### Dysploidy arising from fusion/fission contributes to chromosome polymorphism

To elucidate the factors contributing to variable chromosome numbers in both haploid and diploid isolates, we investigated the collinearity between the chromosomes of the five isolates. To this end, the JP19 and JP8 genomes were first assigned to two subgenomes. Alignment of the genome sequences of JP19 or JP8 with the chromosomes from the three haploids revealed a consistent pattern whereby JP19 and JP8 displayed significantly higher coverage on one chromosome than on the other for each homoeologous chromosome pair (Fig. 3D, E). As a result, we designated the 43 chromosomes of JP19 as subgenome A (JP19$^A$, 22 chromosomes) and subgenome B (JP19$^B$, 21 chromosomes) and the chromosomes of JP8 as subgenome A (JP8$^A$, 21 chromosomes) and subgenome B (JP8$^B$, 20 chromosomes).

Further investigation of the genomes identified repeated interruptions in well-conserved syntenic regions caused by large-scale rearrangements. These rearrangements involved chromosomal fission, fusion, and segmental inversions and translocations, occurring at both inter- and intra-genomic levels (Fig. 3F). Notably, Chr 1 in JP44 is particularly distinguishable because it originates from the fusion of chromosomes syntenic to Chr 1, Chr 2, and a partial segment of Chr 6 from R22 (Figs. 3F and S2). Homoeologous chromosome pairs exhibit high syntenic conservation with minimal rearrangements. More specifically, Chr 17 and Chr 19 from JP8$^B$ likely underwent fusion subsequent to genome hybridization, because their corresponding syntenic chromosomal regions in the three haploids carried two distinct chromosomes. However, this pattern did not occur in the JP19. Overall, it appears likely that fusion/fission and diploidization are important sources of chromosomal polymorphism.

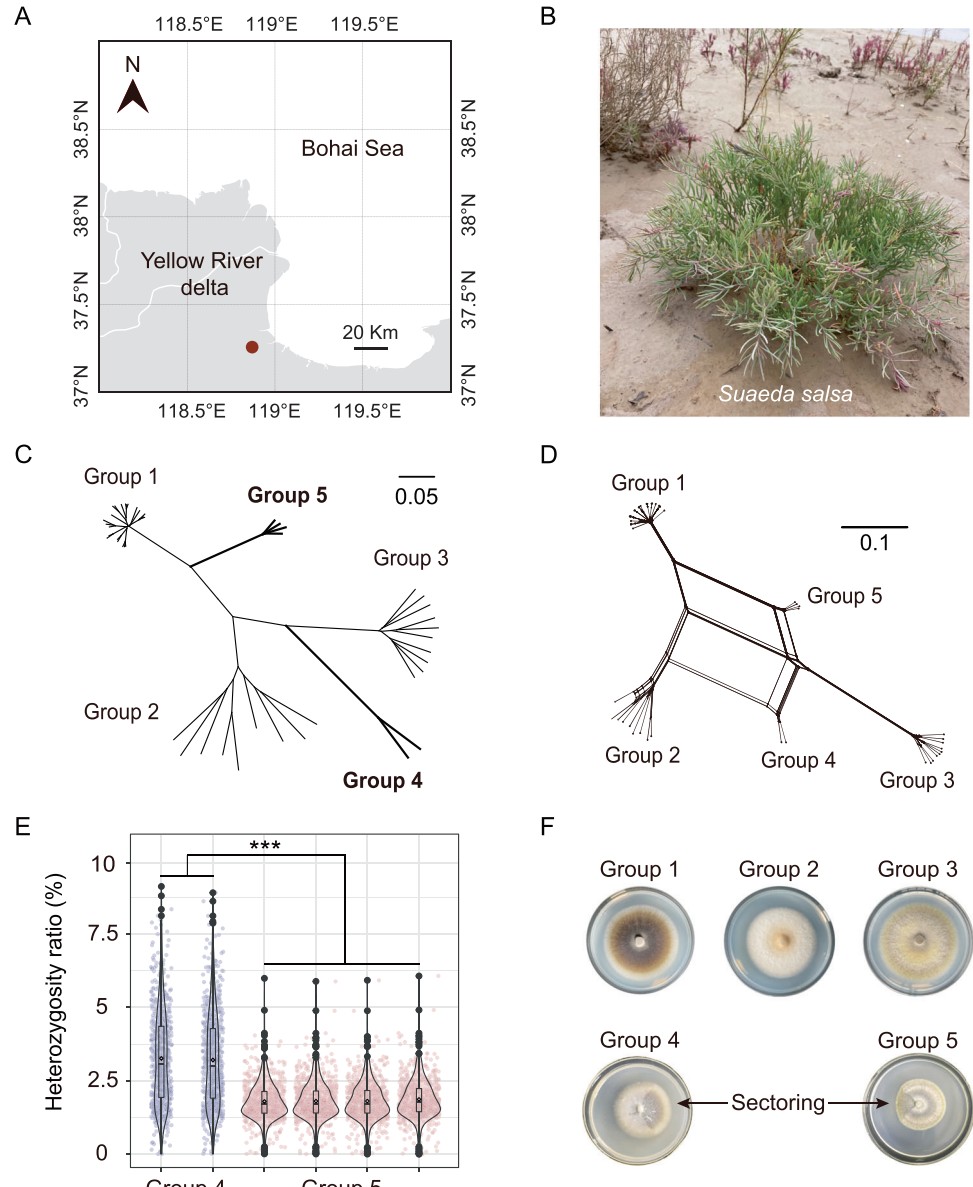

**Fig. 1 | Population structure of *L. rhizohalophila* and the signs of parasexual hybridization. A** The map highlights the sample site in the coastal area of Dongying, Shandong Province, China, marked with a red circle. **B** Growth status of *Suaeda salsa* in natural salt marsh settings. **C** The genetic structure of the *L. rhizohalophila* population. To maintain brevity, isolated names are omitted. The neighbor-joining phylogenetic tree was constructed using 804,906 SNPs, and the scale bar indicates the divergence distance. Heterozygous single nucleotide polymorphisms (SNPs) were converted into degenerate bases (e.g., K = G or T) according to IUPAC, and missing data were coded as N. **D** An unrooted neighbor-net network, generated using the SplitsTree method, illustrates the relationships among the 47 *L. rhizohalophila* isolates. Scale bar represents one nucleotide substitution per site and branch lengths reflect pairwise Hamming distances (uncorrected *p* distances). **E** Heterozygosity measurements and comparisons between Groups 4 and 5 are shown. Data are mean ± SD (*n* = 635 biologically independent samples). Asterisks indicate significant differences between two groups (*t* = 29.973, *p* < 2.2E−16) using a two-sided *t*-test (***\*p* < 0.001). The center of the box-plot represents the median heart rate. The bounds of the box encompass the interquartile range (IQR = Q1–Q3), while the whiskers extend up to 1.5 × IQR units beyond the box boundaries. The open diamonds represent the mean values. Values outside of the upper and lower limits are highlighted as outliers. **F** Sectoring, as indicated by arrows, was occasionally observed in Groups 4 and 5, resembling the characteristics of parasexual haploidization. Source data are provided as a Source Data file.

The *L. rhizohalophila* genomes also exhibited a notable abundance of TEs, constituting approximately 41–48% of the genome content. Among these TEs, long terminal repeat (LTR) retrotransposons, such as LTR/copia and LTR/gypsy, are particularly prevalent (Fig. S3 and Table S1). The distribution of the TEs varied among the seven haplotypes (Fig. 3G), but we found no evidence that the candidate effector genes tended to be associated with gene-sparse and TE-rich regions, indicative of a "one-speed" genome structure (Fig. S4). Furthermore, we identified unique and overlapping gene families across seven haplotype genomes

(Fig. 3H). Specifically, all haplotype genomes shared the highest number of gene families (10,202). We further found that the number of specific gene families (970) unique to Clade 2 was more than 1.8-fold higher than that obtained from Clade 1 (Fig. 4A), thereby indicating potential clade-specific traits arising from evolutionary novelty.

**Pinpointing the diploid formation via parasexual hybridization**
Our subsequent investigation aimed to uncover the evolutionary origins of diploid isolates, as tracing potential parents is crucial for studying

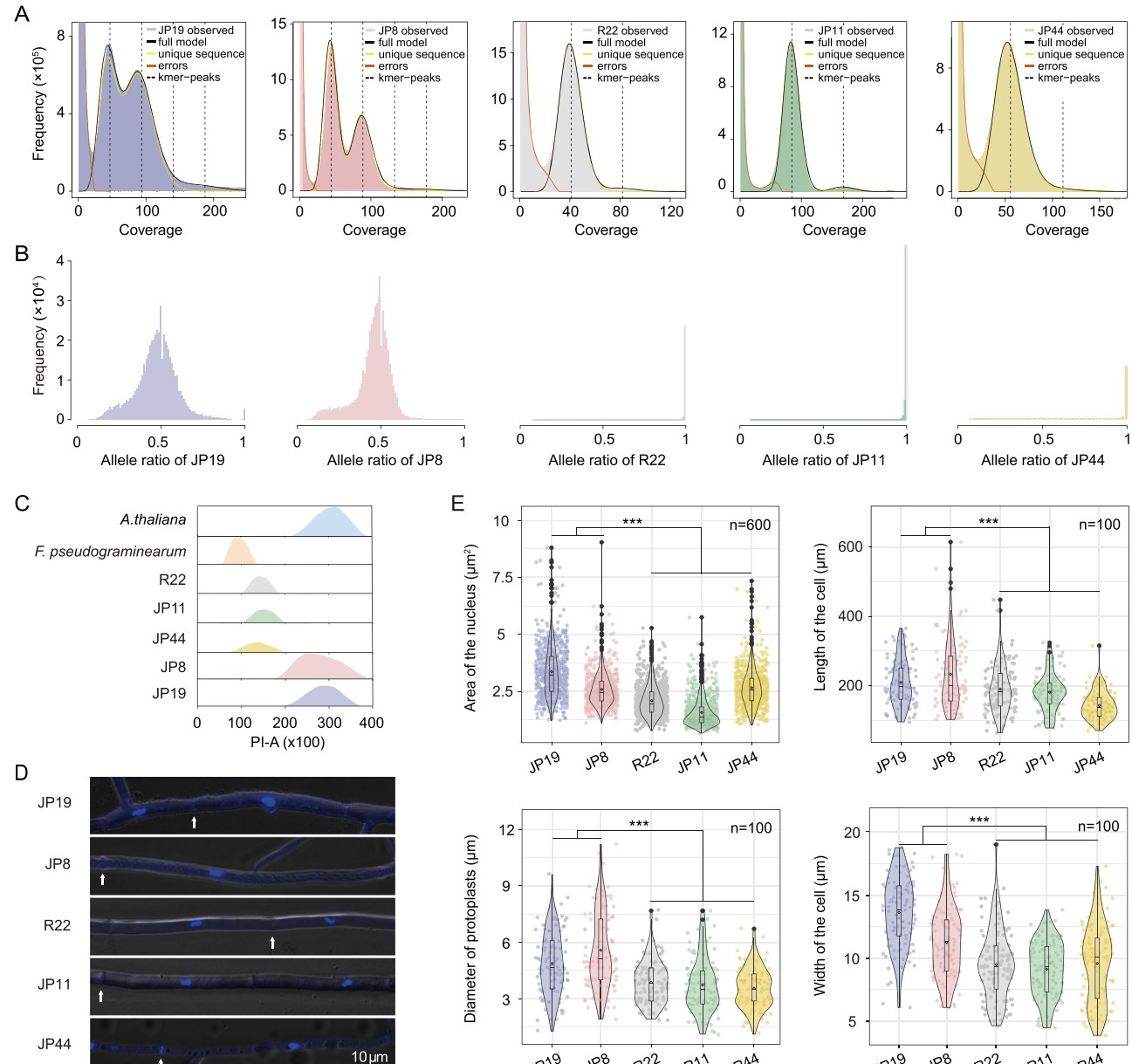

**Fig. 2 | Genomic and cytogenetic evidence of hybridization-mediated genome doubling. A** GenomeScope plots for JP19 and JP8 exhibited a bimodal distribution in the 17-mer frequency, with one peak at approximately 100× (representing homozygous regions) and another peak at approximately 50× (representing heterozygous regions). One peak in the frequency of unique 17-mers was observed within the sequencing data of three haploids (R22, JP11, and JP44), indicating the absence of heterozygous loci. **B** The allele frequency plot displays single nucleotide polymorphisms (SNPs) at an allele frequency of 0.5 for two diploids and 1 for three haploids. **C** Flow cytometry was used to estimate the genome sizes of the five isolates. **D** Nuclei from the five isolates were stained with 4', 6-diamidino-2-phénylindole (DAPI), and hyphal septa are indicated by white arrows. Images shown are representatives from experiments performed in triplicate. **E** Significant

differences between diploid and haploid isolates are revealed in nuclear size ($\chi^2 = 558.36$, $p < 2.2\text{E}{-}16$), protoplast size ($\chi^2 = 86.84$, $p < 2.2\text{E}{-}16$), as well as the length ($F = 57.38$, $p = 1.76\text{E}{-}13$) and width ($\chi^2 = 104.48$, $p < 2.2\text{E}{-}16$) of hyphal cells, as determined by the non-parametric Kruskal–Wallis test and one-way analysis of variance (ANOVA) analyze. Asterisks indicate significant differences (***$p < 0.001$). Data are mean ± SD (Each $n$ number was biologically independent samples). The center of the box-plot represents the median heart rate. The bounds of the box encompass the interquartile range (IQR = Q1–Q3), while the whiskers extend up to 1.5 × IQR units beyond the box boundaries. The open diamonds represent the mean values. Values outside of the upper and lower limits are highlighted as outliers. All the experiments were repeated three times with similar results. Source data are provided as a Source Data file.

genomic and transcriptomic evolution following hybridization[16]. To elucidate the evolutionary relationships among the seven haplotype genomes, we utilized 7085 single-copy genes to infer phylogeny (Fig. 4A). The genomes of R22, JP44, JP19[A], and JP8[A] clustered together to form a monophyletic clade, whereas those of JP11, JP19[B], and JP8[B] formed a distinct clade. This clade supports the hypothesis that the two JP19 subgenomes are genetically closely related to R22 and JP11. Moreover, JP8[A] was found to have a close relationship with JP44,

although the exact progenitor of JP8[B] remains unknown. JP11 is a possible extant relative, although this is not a perfect match. Recent sampling and sequencing of additional 114 isolates revealed that none of their genomes matched that of JP8[B] (SRA accession numbers: SRR26626062-SRR26626175), suggesting that one of the progenitors of JP8 may be extinct due to physiological dysfunction[17]. These findings indicate that JP19 and JP8 likely had independent origins and divergent evolutionary trajectories.

**Table 1 | Summary of genome assemblies and annotations of five *L. rhizohalophila* isolates**

| Values for assembly | R22 | JP44 | JP11 | JP8 | JP19 |
|---|---|---|---|---|---|
| Total genome size (Mb) | 64.01 | 60.93 | 69.56 | 129.52 | 131.87 |
| Total sequenced bases of HiFi (Gb) | 8.27 | 40.85 | 8.76 | 17.16 | 91.46 |
| Sequence depth of HiFi | 129.2 | 669.65 | 125.93 | 132.49 | 693.35 |
| Total sequenced bases of Hi-C (Gb) | 46.54 | 40.68 | 9.21 | 20.25 | 36.73 |
| Sequence depth of Hi-C | 727.07 | 666.88 | 132.4 | 156.35 | 278.53 |
| Percentage of anchoring (%) | 99.55 | 98.68 | 98.24 | 98.19 | 98.61 |
| N50 scaffold length (Mb) | 2.84 | 3.04 | 3.26 | 3.32 | 2.89 |
| Chromosome number | 22 | 21 | 21 | 41 | 43 |
| Telomere sequences at both ends | 19 | 15 | 17 | 23 | 33 |
| Telomere sequences at one end | 3 | 5 | 4 | 16 | 10 |
| Total gene number | 13,540 | 13,404 | 15,626 | 28,792 | 28,679 |
| Average gene length (bp) | 1572 | 1588 | 1549 | 1561 | 1558 |
| BUSCO completeness (%) (S/D)[a] | 99.2 (98.8:0.4) | 98.9 (98.4:0.5) | 99.1 (98.3:0.8) | 98.8 (2.2:96.6) | 98.7 (2.8:95.9) |

[a]S = complete and single-copy BUSCOs, D = complete and duplicated BUSCOs.

As mentioned earlier, we conducted tests to investigate whether the heterozygous SNPs in JP19 and JP8 arose from an admixture between the haploid lineages. For this analysis, we focused only on SNPs from single-copy genes, which are commonly regarded as evidence for hybridization events[18]. Through the inference of biallelic SNP origins, we discovered that hybrid genotypes with heterozygous profiles did not perfectly match the two haploid groups. This suggests that these hybrid genotypes are not F1 hybrids, but could instead stem from the backcrossing of hybrids with other genotypes or result from additional hybrid events. Several nuclear alleles were unique, indicating that hybridization increased genetic variability within the population (Fig. 4B). PCR analysis of the selected loci confirmed this finding (Fig. 4C). Genome-wide pairwise identities for all pairs of haplotype genomes also support genetic relatedness (Fig. 4D). Based on these observations, JP19 and JP8 could be considered as intraspecific diploid hybrids.

**One stable diploid displays the conditional growth vigor**
Polyploidy, characterized by the presence of multiple sets of chromosomes, is often associated with fixed heterosis, a phenomenon in which the hybrid exhibits increased vigor compared with its haploid parents. Based on this assumption, we hypothesized that the two diploid isolates may demonstrate varying degrees of hybrid vigor compared with their corresponding haploid parents. To test this hypothesis, we conducted quantitative phenotypic analyses to measure fungal growth response to a wide range of abiotic stressors in vitro. JP19 consistently exhibited growth fitness advantages over the haploids in all cases (Fig. 5). However, these advantages were condition-dependent and were not apparent under higher levels of abiotic stress (Fig. S5). Specifically, JP19 demonstrated growth vigor under high levels of oxidative stress, with optimal mycelial growth observed under conditions of salinity and alkalinity (Fig. 5A, B). In contrast, JP8 displayed susceptibility to all abiotic stressors and exhibited intermediate traits between those of the parents, indicating a decrease in susceptibility. These findings suggest that hybridization events can coincide with the emergence of metabolic and physiological innovations or the dysregulation of abiotic stress homeostasis. Additionally, JP19 exhibited transgression of eight physiological traits relative to its haploid parents (Fig. 5C–H). When we expanded the sample group to include all individuals from groups 2 and 3, it became evident that JP19 growth still consistently surpassed that of all included haploids under the described stressful conditions (Fig. S6).

**Transcriptome atlas and manifestation of subgenome dominance**
Subgenome dominance is characterized by differences in gene expression and TE loads between subgenomes[19]. Our findings revealed that subgenome B had a larger size, higher gene content, and higher TE density than subgenome A in both JP19 and JP8 (Table S2). To identify signs of subgenomic dominance, we compared the expression levels of the homologous genes. The transcriptomes were sequenced in vitro and *in planta* mycelia (Fig. 6A). Principal component analysis (PCA) showed distinct separation of the haploids from clade 1 (JP11), clade 2 (R22 and JP44), and the two diploids along the PC1 axis, with the hybrids occupying intermediate positions between the haploid lineages, further supporting their evolutionary relationships and hybridization outcomes. Additionally, the transcriptomes of the hyphae grown in vitro and *in planta* were clearly distinguishable along the PC2 axis.

Based on the expression levels of the 10,731 and 10,662 homoeologous gene pairs in JP19 and JP8, respectively (Fig. 6B), we compared the genome-wide transcriptional levels of subgenomes A and B. The majority of homoeologous gene pairs (67.4–70.9%) exhibited unbiased expression, indicating that the transcriptional shock induced by hybridization may be limited (Fig. 6B). However, we identified a subset of expressed homoeologous gene pairs with a greater than 2-fold difference in expression between subgenomes A and B across the four sample types. Specifically, in JP19, homoeologs displayed higher expression in subgenome A than in subgenome B under all tested conditions, and these differences were either significant or borderline significant. Approximately 11–13% of the genes exhibited differential expression between subgenome A and B homoeologs. In contrast, in JP8, no significant global homoeolog expression bias toward a specific subgenome was observed under all in vitro growth conditions, except for higher homoeologous expression in subgenome A under in planta growth conditions (Fig. 6B). In summary, our results indicate a bias toward subgenome A in terms of global expression level dominance in JP19, but the impact of genomic shock is largely buffered.

**Additive and nonadditive gene expression patterns across four growth conditions**
We conducted further investigations to track changes in gene expression and explore the diploidization process. Comparisons were made between JP19 and JP8 and their respective parents to identify transcriptomic alterations and uncover the molecular mechanisms underlying the enhanced fitness of JP19. Through comparative

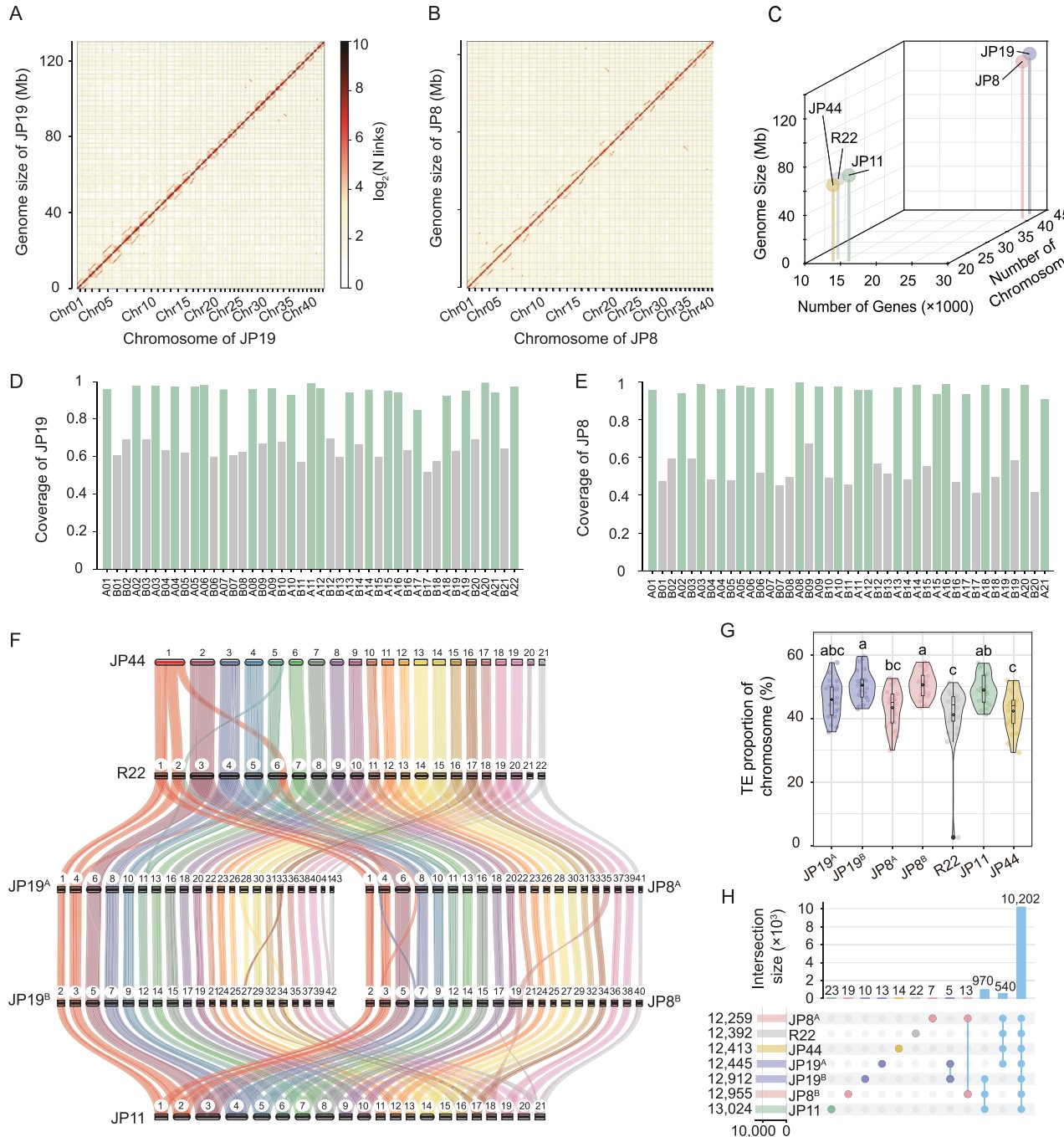

**Fig. 3 | Unveiling the presence of chromosomal polymorphisms in *L. rhizohalophila*. A**, **B** Hi-C interaction frequency distributions among chromosomes for the two diploid hybrids are presented via genome-wide interaction heatmaps (at 40 kb resolution). The log$_2$ transformation of the link number highlighted distinguishable chromosome groupings. The heatmap color key, which varies from light yellow to dark red, represents the frequency of the Hi-C interactions. **C** JP19 and JP8 exhibit nearly doubled genome sizes, gene numbers, and chromosome numbers compared to R22, JP44, and JP11. **D**, **E** Subgenome assignments are made, as evidenced by coverage histograms showing the alignment of JP19 and JP8 genomic sequences to R22 and JP44 chromosomes, respectively. Chromosome IDs were reassigned to represent the two sets of homoeologous chromosomes. **F** Chromosomal structure and collinearity among the seven haplotype genomes, with collinear regions between chromosomes connected by ribbons of different colors. Chromosome comparisons showed perfect collinearity, except for some major rearrangement

events. The differences in chromosome numbers were due to a few chromosome fusions and fissures. The chromosome numbers are indicated at the top of each chromosome. **G** Violin plot shows a comparison of the TE composition among the seven haplotype genomes ($\chi^2 = 40.00$, $p = 4.54e{-}07$) by non-parametric Kruskal–Wallis test. The TEs in JP19$^B$ and JP8$^B$ were significantly higher than those in R22 and JP44. Data are mean ± SD ($n = 22$ chromosomes of JP19$^A$, and R22, $n = 21$ chromosomes of JP19$^B$, JP8$^A$, JP11, and JP44 and $n = 20$ chromosomes of JP8$^B$). The center of the box-plot represents the median heart rate. The bounds of the box encompass the interquartile range (IQR = Q1–Q3), while the whiskers extend up to 1.5 × IQR units beyond the box boundaries. The open diamonds represent the mean values. Values outside of the upper and lower limits are highlighted as outliers. Different letters indicate statistical significance at $p < 0.05$, as determined by Conover's test. **H** UpSet diagram illustrating shared and unique orthologous gene families across all seven haplotypes. Source data are provided as a Source Data file.

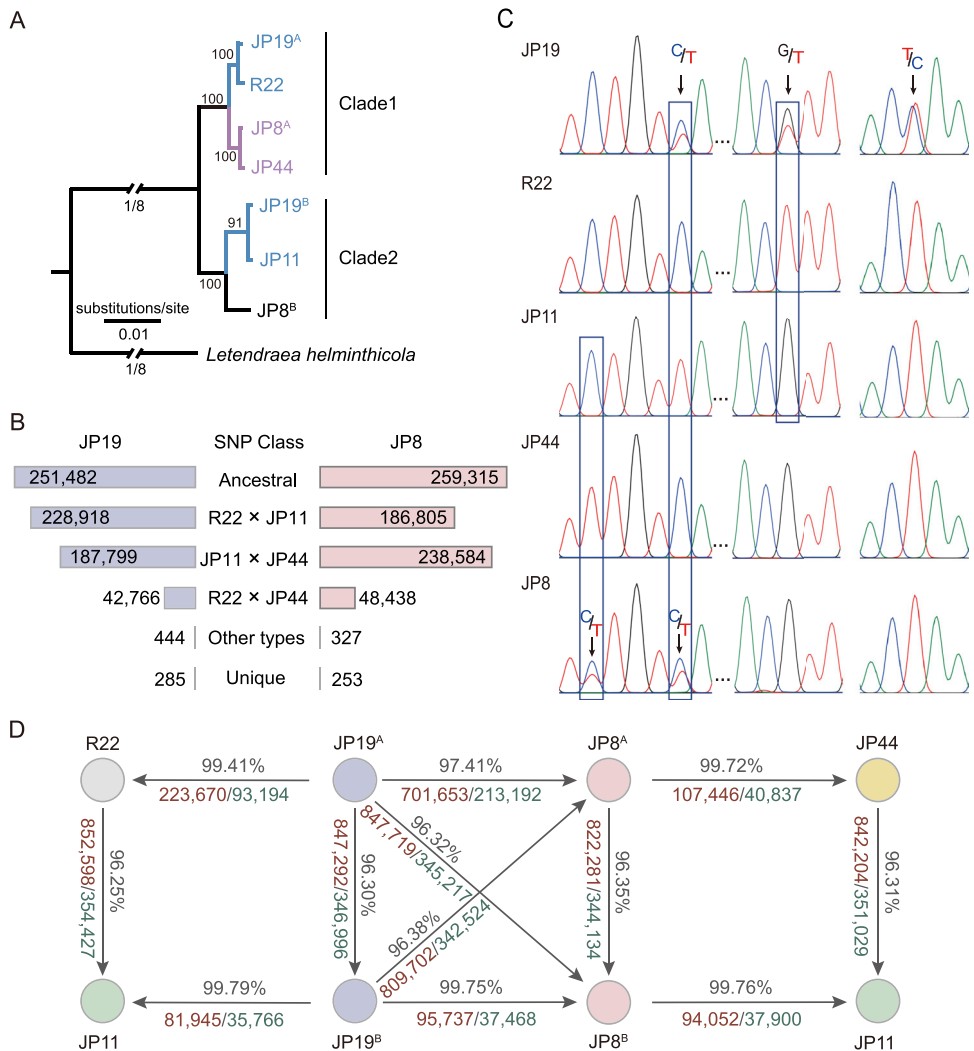

**Fig. 4 | Origin of progenitors for the diploid hybrids JP19 and JP8.**
**A** Phylogenetic tree based on orthologous genes derived from the seven haplotype genomes. Branch lengths indicate estimated nucleotide substitutions per site.
**B** Single nucleotide polymorphism (SNP) counts by diagnostic class were analyzed for the JP19 and JP8 isolates. The presence of extensive allelic variation in hybrid genotypes, including novel nuclear alleles, suggests that hybridization has enriched the genetic variability in the population or potentially involves a third parent. We further deduced the origin of bi-allelic SNPs, with JP19 primarily originating from R22 and JP11, while JP8 mainly originated from JP44 and JP11. This broadly corresponded to the observed phylogenetic relationships. Notably, most sequences did not perfectly match the pairwise combinations of the three parental haploids. **C** The presence of heterozygous SNPs was validated by PCR amplification and Sanger sequencing. The JP19 and JP8 isolates exhibited two alleles at each locus examined (indicated by arrows), whereas JP44, JP11, and R22 possessed only a single allele. **D** Sequence identity, number of SNPs, and indels between R22, JP11, and JP44, and the subgenomes of JP19 and JP8 were compared. Text marked in black, red, and green represents sequence identity, SNPs, and Indels, respectively.

transcriptomic analysis of each hybrid parent triad, we observed a relatively high percentage of differentially expressed genes. These expressed genes could be categorized into eight groups, which is consistent with previous reports[20]. A total of 12,754 and 12,476 genes were expressed in JP19 and JP8, respectively. Our focus was primarily centered on expression patterns where the total expression level of homeologs increased or deviated from the mid-parental level (non-additivity). The alluvial plot in Fig. 6C illustrates the changes in expression patterns observed in JP19 across the four growth conditions.

Overall, the general trend indicated that a large proportion of JP19 homeologs (ranging from 64.1% to 83.8%) displayed no differential expression compared with their haploid orthologs (additivity) across the four growth conditions, indicating that most genes did not show differential expression patterns. However, we did observe a fraction of expressed genes (35.9%) that exhibited nonadditive expression, with an increase in the number of transgressively upregulated (TUR) genes

in JP19 under 0.3 M NaCl compared to other growth conditions. In this case, transgressive expression was observed for 2275 genes (17. 5%) in JP19, with downregulation similar to upregulation. Our results further illustrated that *cis-* effects played a more significant role than *trans-* effects in shaping differential homeolog expression patterns (Fig. S7). Additionally, the number of TUR genes in JP18 (741) was much lower than that found in JP19 (1249) at 0.3 M NaCl. Detailed categories of additive and nonadditive gene expression patterns in JP8 are shown in Fig. S8.

## TUR genes are enriched in functions associated with membrane phospholipid biosynthesis

To gain comprehensive insights into the mechanism underlying the enhanced growth of JP19 when exposed to salt treatment (0.3 M NaCl), we focused on genes displaying TUR, which could serve as potential candidates for heterosis[21]. TUR genes can be further categorized into two types based on non-differential (NDE) and differential (DE)

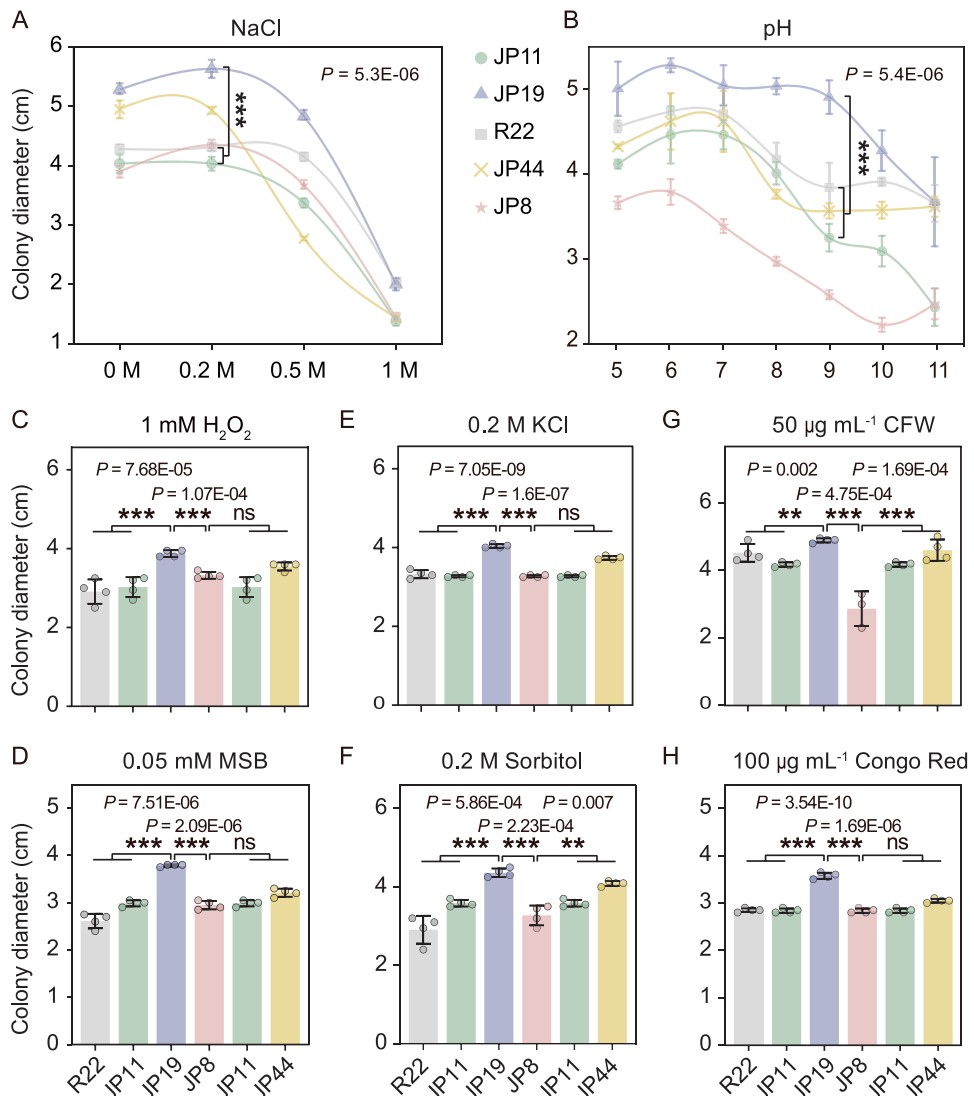

**Fig. 5 | Growth vigor of JP19 when exposed to a wide range of abiotic stressors.** **A**, **B** Fungal growth under saline-alkali stress conditions (NaCl and pH). Data are mean ± SD ($n = 5$ biologically independent samples). **C**, **D** The impact of oxidative stress ($H_2O_2$ and MSB) on growth was evaluated. **E**, **F** Growth performance was analyzed under osmotic stress conditions (KCl and sorbitol). **G**, **H** Response to cell wall stress (CFW and Congo red) was examined. Data were subjected to an one-way analysis of variance (ANOVA). Data are mean ± SD for (**C**–**H**) ($n = 4$ biologically independent samples). Asterisks indicate significant differences between diploids and the corresponding parent haploid isolates, as determined by Duncan's test at significance levels of $p < 0.05$ (*** $p < 0.001$, ** $p < 0.01$, ns: $p \geq 0.05$). All the data were repeated three times with similar results. MSB 2-methyl-1,4-naphthoquinone, CFW calcofluor-white. Source data are provided as a Source Data file.

expression levels between the two parents. Our findings revealed that the 864 NDE-TUR genes were significantly enriched in 19 GO terms in the biological process (BP) category ($p < 0.05$), and the over-representation was caused specifically by the enrichment of GO terms "lipid metabolic process" (GO:0006629, the most enriched term), "glycerolipid metabolic process" (GO:0046486), "lipid localization" (GO:0010876), and "cellular lipid metabolic process" (GO:0044255) (Fig. 7A, B, Table S3 and Supplementary Data 1). These terms included genes encoding synthases and other genes previously reported to participate in phospholipid biosynthesis (see more details in Fig. 8), and these candidate genes with transgressive upregulation contribute to the membrane integrity and fluidity of fungi under salinity stress. In addition, other enriched GO terms were also associated with stress response, including "cation transport" (GO:0006812, 25 genes), "cellular response to abiotic stimulus" (GO:0071214, 18 genes), and "stress-activated MAPK cascade" (GO:0051403, 7 genes) (Tables S3 and S4). On the contrast, the 385 DE-TUR genes were only categorized into two GO terms "heme binding" (GO:0020037) and "tetrapyrrole binding"

(GO:0046906), whereas their roles in response to abiotic stress were still ambiguous. However, no significant GO terms related to the response to salinity were identified for the TUR genes in JP19 under the other three growth conditions. Similarly, among the TUR genes in JP8 across all growth conditions, no significant functional enrichment in any class of GO terms with key roles in salinity tolerance was detected. The results of a series of in vitro assays were consistent with the inference that JP19 maintains active membrane lipid metabolism when exposed to optimal saline stress. As expected, minimal staining with PI was observed in JP19 mycelia, indicating limited disturbance of membrane integrity under 0.3 M NaCl (Fig. S9). Additionally, an increased abundance of mycelial lipid droplets was observed in 0.3 M NaCl ($F = 65.74$, $p < 0.001$) (Fig. 7C, D), which may contribute to membrane phospholipid biosynthesis in response to external stresses[22]. Furthermore, the mycelial malondialdehyde (MDA) content, a product of lipid peroxidation, was significantly reduced in JP19 under 0.3 M NaCl ($F = 9.24$, $p < 0.05$) (Fig. 7E). Collectively, these observations support the notion that JP19 exhibits greater robustness in maintaining active

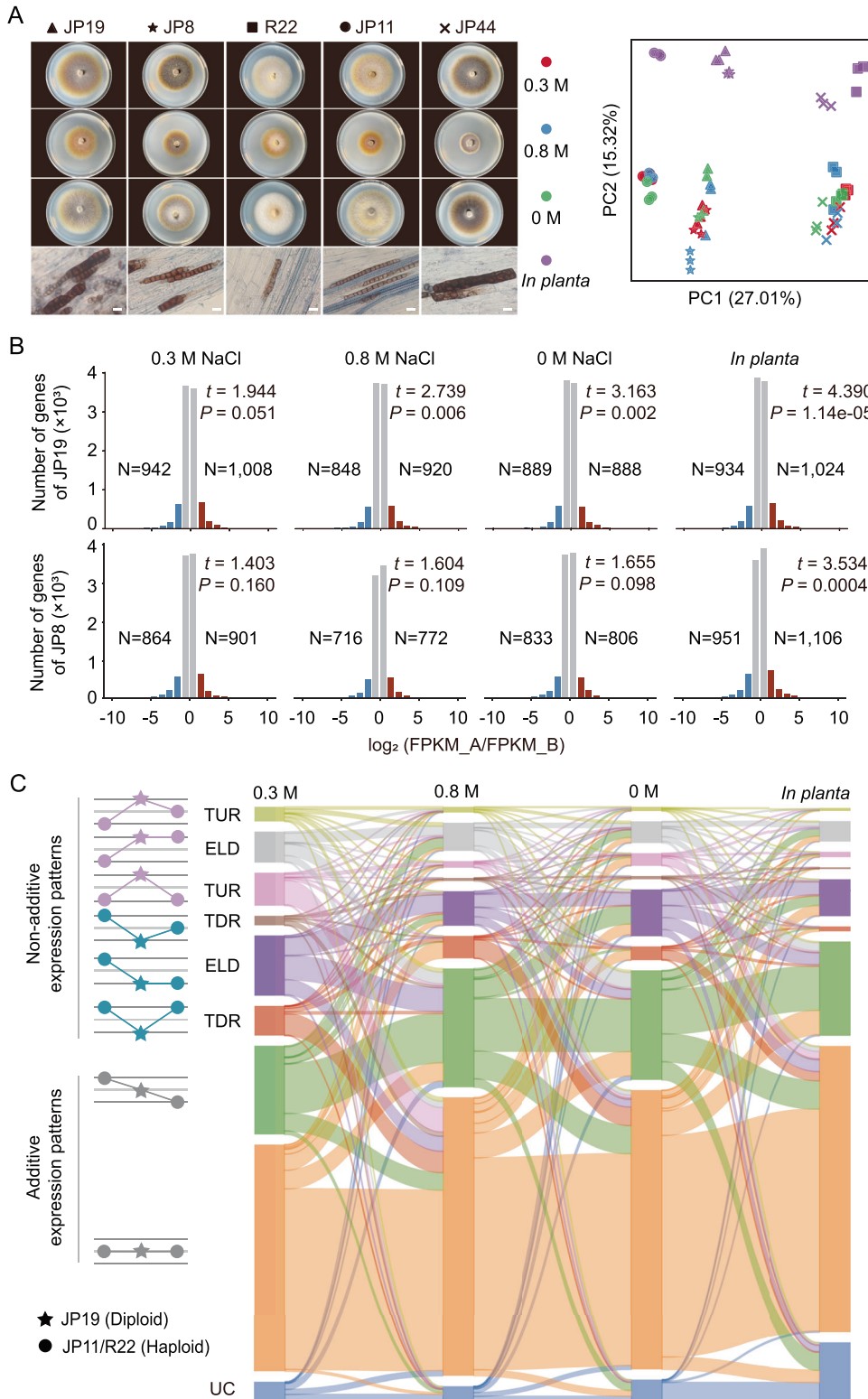

membrane lipid metabolism and transport systems in response to salinity stress.

## Discussion

### A high degree of genome plasticity of *L. rhizohalophila*

Our comprehensive genomic analyses revealed that the sympatric diverged population of *L. rhizohalophila* comprised a mixture of haploid, dysploid, and diploid mosaic lineages. This population exhibited elevated levels of genetic diversity, admixture, and heterozygosity, indicating a remarkable degree of genomic plasticity. Chromosomal reshuffling plays a crucial role in generating diversity in predominantly asexual fungal pathogens, thereby avoiding evolutionary dead ends[23,24]. This phenomenon occurs in the absence of sexual reproduction. Chromosomal imbalances can arise through parasexual activities[25]. Although the gene sequences and synteny remained conserved among the different isolates, the genome of *L. rhizohalophila* displayed extensive fission, fusion, and diploidization events, leading to chromosomal polymorphisms. This challenges the

**Fig. 6 | Gene expression patterns of the diploid hybrid JP19 and subgenome dominance. A** Principal component analysis (PCA) of fungal transcriptomic data from 60 libraries, with each dot representing an individual transcriptome under four different growth conditions (in vitro and *in planta*). Examples of fungal colonies and root infections are also presented. Scale bars = 20 μm. **B** A global bias toward subgenome A in terms of expression level dominance was observed, indicating a certain level of dominance in one subgenome. However, the effects of genomic shocks were largely mitigated. The degree of subgenome dominance was assessed by comparing the expression levels of homologous gene pairs from the two subgenomes. $\log_2(\text{FPKM\_A}/\text{FPKM\_B})$ indicates the magnitude of the expression differences between homoeologous gene pairs. *N* values indicate the number of dominant genes in subgenomes A and B. Differential expression significance was determined using two-sided *t*-tests, with a significance threshold of 0.05. Data are mean ± SD ($n = 13{,}756$ genes of JP19 subgenome, and $n = 13{,}643$ genes of JP8 subgenome). **C** An alluvial plot illustrating the various categories of additive and nonadditive gene expression patterns in JP19 as well as changes in the number of homoeologous genes across the four growth conditions. JP19 had a significant increase in TUR genes under 0.3 M NaCl. ELD expression level dominance, TDR transgressive downregulation, TUR transgressive upregulation, UC unclassified. Source data are provided as a Source Data file.

prevailing assumption that chromosome number and arrangement are generally conserved within individuals of the same species[26]. Notably, the chromosome numbers of haploid *L. rhizohalophila* surpassed those of most ascomycetous lineages[27,28]. It has been demonstrated that varying karyotypic configurations and dysploidy, which can be advantageous in stressed populations[29,30], can result in significant alterations in global gene expression owing to the presence of extra or fewer copies of chromosomes.

## Promiscuous and recurrent hybridizations within the *L. rhizohalophila* population

The occurrence of fungal hybrids has been increasingly reported in recent years, particularly among plant and human pathogens, extremotolerant black yeasts, and plant-beneficial mycobionts[12,31–34]. However, the identity of these hybrids in natural salt marshes remains largely unknown. Our data suggest that diploid *L. rhizohalophila* from halophyte roots likely forms through parasexual hybridization, which is similar to the frequent occurrence of hybrids in grass endophytes[35]. Nevertheless, it is likely that diploid *L. rhizohalophila* hybrids have not been abundant recently, aligning with the general understanding that stable diploids are uncommon in natural environments[36,37].

Several potential scenarios may explain the diploid formation observed in *L. rhizohalophila*. The allelic profiles of the hybrids may have originated from a single hybridization event followed by inbreeding or gene conversion, or from two or more independent hybridization events. Additionally, there were a large number of unique heterozygous SNPs in each group, suggesting the presence of an unknown parent genome. This could explain why the origins of some donor alleles in putative hybrids could not be determined. It is also possible that current genomic data may not provide sufficient resolution to fully uncover multiple independent hybridization events. Our findings align with observations of a crop hybrid pathogen resulting from a cross between *B. graminis* f. sp. *tritici* and *B. graminis* f. sp. *secalis*[38] as well as the grass endophytic fungus *Neotyphodium coenophialum*, which involves three ancestors in two separate hybridization events[39]. Additionally, *L. rhizohalophila* is a homothallic fungus[10], and such a sexual system could limit its potential for recombination[40]. However, our population data point to frequent recombination within the *L. rhizohalophila* population, supporting the idea that parasexual reproduction accompanied by mitotic recombination can confer genetic diversity similar to conventional sexual reproduction[30].

## Diploidization of *L rhizohalophila*: a strategy for saline adaptation?

Compared to homokaryotic relatives, heterokaryotic isolates are often well adapted to fluctuating environmental conditions through the improvement of many life-history traits[41,42]. Additionally, hybridization offers several key benefits including heterosis (hybrid vigor) and homeostasis (robustness to environmental changes)[43]. The evolutionary consequences of changes in ploidy in fungal hybrids following hybridization, including their impact on gene expression, have been extensively studied. However, the strength of heterosis in fungal hybrids remains debatable. It has been suggested that diploids may have an advantage over haploids[44], as two copies of each gene can help purge deleterious mutations through processes, such as the Müller ratchet.

In the context of plant fungal pathogens, hybrids can contribute to increased pathogenicity or enable access to new hosts[35,45,46]. For instance, the allodiploid *Verticillium longisporum* exhibited greater virulence than its haploid relatives[47,48]. Endophytic *Epichloë* allopolyploids produce a diverse range of alkaloids[49], and the diploid hybrid *Coniochaeta* is known for its exceptional lignocellulolytic capabilities[50]. However, only a few studies have directly addressed the assumption of increased fitness in hybrids. It has been shown that there are no substantial physiological differences between haploid and diploid isolates[31], and in some cases, diploidization can even lead to decreased fitness[51].

The significance of ploidy changes in the evolution and physiology of *L. rhizohalophila* in natural salt marshes is intriguing given the various abiotic stresses that these organisms encounter. While JP19 displays several advantages over parental haploids in terms of growth under various stresses, the lack of growth vigor in JP8 is perplexing. One possible explanation could be its lower heterozygosity compared to JP19. Increased heterozygosity is commonly associated with enhanced growth vigor, which is a prerequisite for changes in gene expression and phenotypic variation in allopolyploids. Furthermore, the occasional sectoring observed in JP8 during successive culturing suggests that it may be in the early stages of post-hybridization evolution, experiencing genetic incompatibilities between the subgenomes. Thus, the unstable hybrid genome of JP8 impaired growth fitness. Finally, we observed differential expression patterns of homoeologs between JP19 and JP8. It is evident that homoeolog expression bias is significant in JP19 but not in JP8 under saline conditions. This discrepancy may be attributed to the different epigenetic mechanisms associated with variations in the TEs content. Therefore, it is reasonable to assume that hybrids JP19 and JP8 descended from closely related ancestors but evolved divergent mechanisms.

Another intriguing question that remains unexplained is why only one diploid isolate JP19 exhibits growth vigor (Fig. S10), as in total we identified two lineages (group 4 and group 5) comprising six diploid hybrids. Evolutionarily speaking, the adaptive diploid population should be expanded. Two possible alternative explanations emerge. First, the limited sampling failed to identify more diploids. Second, the stable diploid hybrids associated with heterosis may occur in unique habitats.

## Hybrid heterosis: perspectives from genomic evolution to transcriptional output

From a transcriptomic perspective, the robust resistance to environmental changes and growth vigor observed in hybrids may be attributed to the nonadditive expression of several key genes[52]. However, the mechanism underlying heterosis remains unclear, and efforts to elucidate it have been impeded by a lack of information regarding the ancestral origins of fungal diploids, which hampers further comparisons of their physiological and transcriptional traits.

To date, only two case studies have examined different categories of triad genes in fungal diploids, specifically *Neotyphodium lolii* × *Epichloë typhina* and *V. longisporum*[24,53]. However, these studies did not find any evidence of growth vigor. In contrast, the present study

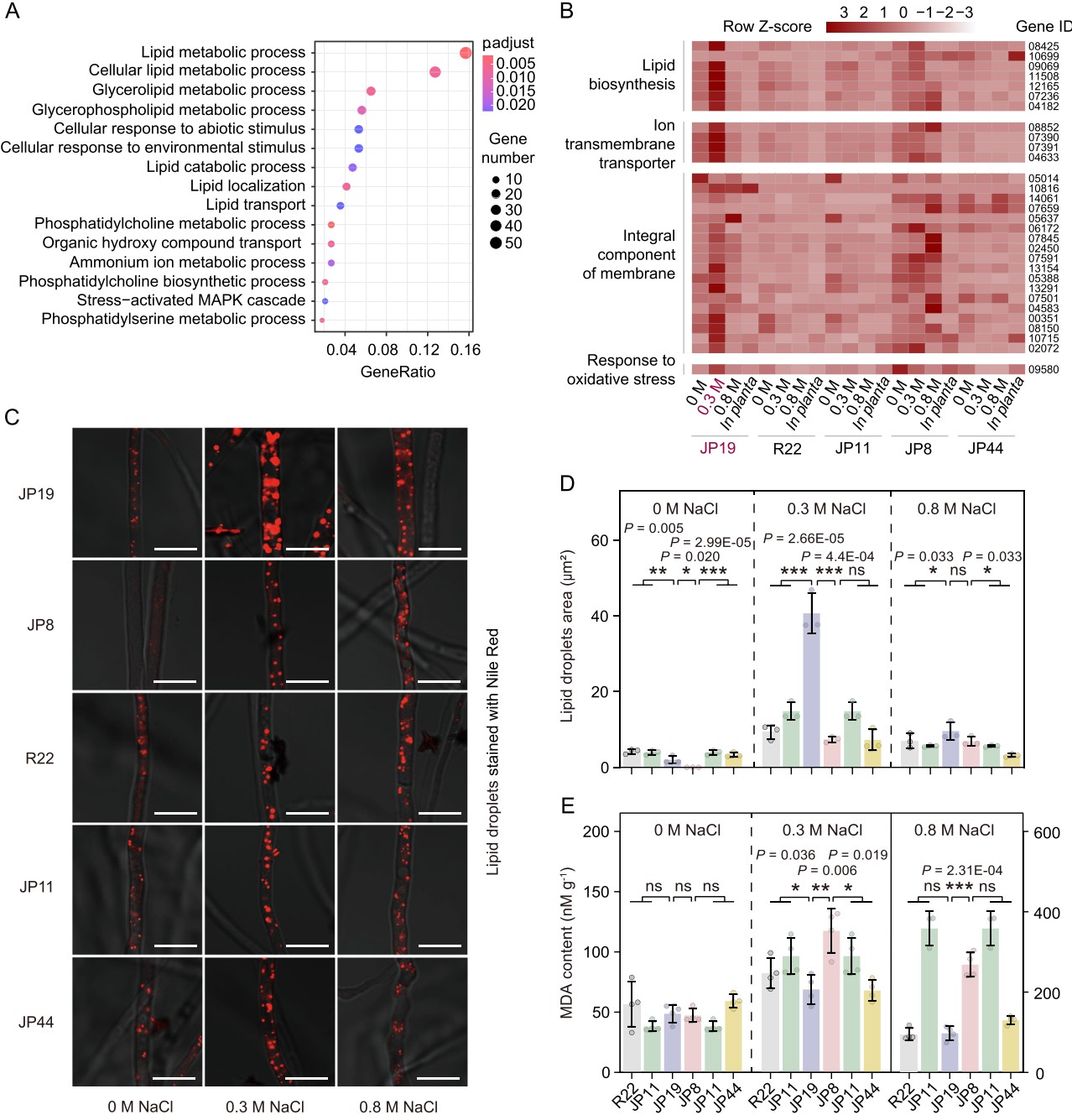

**Fig. 7 | Transgressively upregulated genes enriched in membrane glycerolipid biosynthesis and measurement of membrane lipid function. A** Gene ontology (GO) enrichment analysis and scatter plot of non-differential transgressively upregulated (NDE-TUR) genes in JP19 under 0.3 M NaCl are shown, highlighting the top 15 enriched functions related to membrane glycerolipid biosynthesis. The vertical axis represents the description of the GO terms. The *p* values calculated by one-sided hypergeometric test and corrected for multiple testing using the Benjamini–Hochberg approach. The *p* values (*p*adj) are indicated by a color scale, and the dot size corresponds to the number of TUR genes linked to the respective functions. **B** A heatmap is presented to demonstrate changes in the expression patterns of representative NDE-TUR genes among the five isolates under the four growth conditions. These genes were selected from all the NDE-TUR genes with FPKM > 5 and Log₂FC > 2, some of which are involved in membrane glycerolipid

biosynthetic processes (Fig. 8). **C** Nile red staining was used to visualize mycelial lipid droplets. Numerous clusters of lipid droplets were observed in JP19 under 0.3 M NaCl. Scale bars = 10 μm. Images shown are representatives from experiments performed in triplicate. **D** One-way analysis of variance (ANOVA) was conducted to compare lipid droplet size per hyphal cell among the five isolates. Data are mean ± SD (*n* = 3 biologically independent samples). **E** ANOVA revealed significant differences in malondialdehyde (MDA) production among the five isolates after salt exposure. Data are mean ± SD (*n* = 4 biologically independent samples). Asterisks indicate significant differences between diploids and the corresponding parent haploid isolates, as determined by Duncan's test at significance levels of *p* < 0.05 (***p* < 0.001, ***p* < 0.01, ns: *p* ≥ 0.05). All the experiments were repeated two times with similar results (**D**, **E**). Source data are provided as a Source Data file.

demonstrated the potential for heterosis-related resistance in a diploid hybrid of DSE fungi. We observed a significant proportion of non-additive gene expression in JP19 under 0.3 M NaCl conditions, whereas such expression was largely attenuated in other conditions.

Particularly noteworthy was the concurrent upregulation of several genes involved in lipid production, which likely contributes to membrane integrity and fluidity, consequently enhancing salt tolerance. Although the precise mechanisms underlying these transgressive

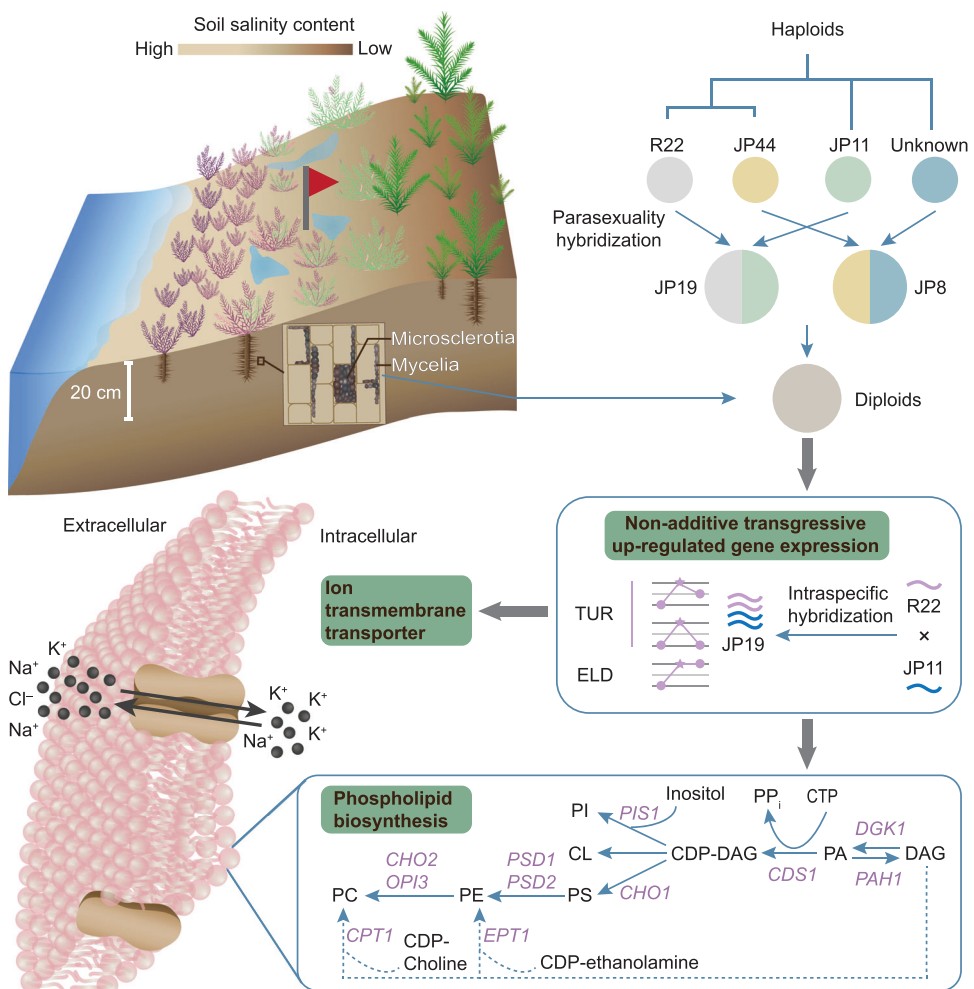

**Fig. 8 | A simplified model illustrates a plausible scenario of diploid hybrid formation and the potential molecular mechanisms that contribute to fungal growth vigor in response to salinity stress.** In this study, we present the most likely evolutionary scenario for diploid origin. Parasexual hybridization between closely related haploid individuals may play a role in diploid formation. Notably, the diploid JP19 exhibited pronounced growth vigor when exposed to various abiotic stressors. While 0.3 M NaCl acts as a stimulus for JP19, it serves as a stressor for the other isolates. A set of nonadditive genes that are primarily involved in membrane phospholipid biosynthesis and ion transmembrane transporters are transgressively upregulated under 0.3 M NaCl, suggesting a key role in maintaining membrane integrity and improving fungal fitness under salinity stress. The related genes were identified as described[91]. Here, we provide abbreviated gene names and their corresponding functions: DGK1 (DAG kinase), PAH1 (phosphatidic acid phosphatase), CDS1 (CDP-DAG synthase), PIS1 (phosphatidylinositol synthase), CHO1 (phosphatidylserine synthase), PSD1 and PSD2 (phosphatidylserine decarboxylases), CHO2 (phosphatidylethanolamine methyltransferase), OPI3 (phospholipid methyltransferase), EPT1 (ethanolamine phosphotransferase), and CPT1 (choline phosphotransferase). CL cardiolipin, PC phosphatidylcholine, PE phosphatidylethanolamine, PI phosphatidylinositol, PS phosphatidylserine.

effects remain largely unknown, they are predicted to involve a combination of *cis* and *trans* factors[54]. Overall, this study provides valuable insights into the transcriptional effects of hybridization in a natural hybrid, leading to the rewiring of gene regulatory networks and reinforcement of metabolic efficiencies.

### Diploid *L. rhizohalophila*: an emerging model for studying fungus-driven plant fitness
Our work further presents a conceptual model for examining the consequences of diploidization on the evolution of fungi that form mutualistic associations with roots. This model has significant implications not only for understanding the dynamics of habitat survival, but also for enhancing the fitness of host plants. However, it is important to note that ascribing improved growth vigor to enhanced host or nonhost plant fitness is premature at this stage. To validate this hypothesis, extensive fungal inoculation experiments using native and model plants in controlled environments and field trials are required. Such experiments will contribute to the development of superior microbial inoculants (if any) that can promote

the growth of trees and crops in salt marsh ecosystems by conferring innovative properties.

In conclusion, this study unveils a novel hybrid form of DSE *L. rhizohalophila* and delves into the factors driving heterosis. Our findings deepen the understanding of the genetics of root-associated mycobionts and their ability to produce hybrids by introducing genomic plasticity and environmental adaptability. Diploidization played a crucial role in the rapid evolution of *L. rhizohalophila*. Moreover, our investigation lays the groundwork for further examination of the fitness landscape of abnormal fungal ploidy levels, particularly in the context of symbiotic interactions with plants. Figure 8 depicts a plausible scenario for the formation of diploid-like isolates and the potential molecular mechanism responsible for enhanced fungal growth under salinity stress.

## Methods
### Isolate collection, culturing and genomic DNA extraction
In the present study, 47 *L. rhizohalophila* isolates were used. This fungus was isolated from the roots of *Suaeda salsa*, which is native to

saline soils in the Yellow River Delta (Dongying City, Shandong Province, China) (Supplementary Data 2). Genomic data and living cultures of 29 individuals were publicly accessible, as previously reported[10]. From the native sampling site, we collected 18 specimens, which are illustrated in Fig. 1A, B. The method for fungal isolation was provided in previous[9]. Briefly, the healthy roots were washed by tap water followed by surface sterilization with an ethanol-NaClO-based treatment. Root segments of 0.5 cm length were placed on potato dextrose agar (PDA) in 9 cm Petri dishes. The dried specimens were deposited in the Herbarium of the Institute of Microbiology, Academia Sinica, under accession numbers HMAS352405-HMAS352422. Mycelial cultures were deposited at the China General Microbiological Culture Collection Center (collection numbers: CGMCC3.24571-CGMCC3.24588). The cetyltrimethylammonium bromide (CTAB) method[55] was used to extract genomic DNA from freshly harvested mycelia. The concentration and purity of the extracted DNA were assessed using a Qubit fluorometer and Nanodrop 2000 spectrophotometer (Thermo Fisher Scientific, Carlsbad, CA, USA). The integrity of the DNA was evaluated using 1% agarose gel electrophoresis.

## Population genomics analysis

To expand our dataset, we collected genomic data from an additional 18 isolates of *L. rhizohalophila*. These individuals were sequenced without prior knowledge of their ploidy status. Genome resequencing was performed using MGISEQ libraries with an insert size of 350 bp. The MGISEQ-2000 instrument was used for paired-end sequencing with a read length of 150 bp (BGI, Wuhan, China) (Table S5). For SNP calling, high-quality DNA sequencing reads which filtered were aligned to JP44 reference genome using the MEM module and BWA v0.7.17[56]. Then SAM alignment files were sorted and converted to BAM files using SAMtools v1.3[57]. Next, The BCFtools v1.10.2 package was used to call, filter, and merge SNPs. The filter parameters for each sample were set as follows: "%QUAL < 20 || DP < 10 || DP > 400". Subsequently, the merged SNPs were further filtered using VCFtools[58] with the parameters "--maf 0.02 --max-missing 0.5 --min-alleles 2 --max-alleles 2." To determine the phylogenetic relationships among individuals, a NJ phylogenetic tree was constructed using TreeBest v1.9.2. The p-distance model based on population-scale SNPs was employed for this analysis. To assess recombination events between groups, potential recombination events in the entire dataset were visualized using SplitsTree v4.14.4 (http://splitstree.org)[59] based on pairwise distances with the Kimura K3ST model. This approach enables the identification of a reticulate evolution rather than a strictly bifurcating path. Interestingly, during the SNP calling process, we observed long stretches of heterozygous SNPs within the two subpopulations, suggesting potential heterokaryon or diploid formation. To quantify this observation, we calculated heterozygosity in non-overlapping sliding windows of 100 kb. Heterozygosity was defined as the proportion of heterozygous SNPs within each window for each individual.

## Genome sequencing, chromosome-level assembly and annotations

The identification of heterozygous SNPs within the *L. rhizohalophila* population prompted us to investigate potential hybridization events and genome evolution. To obtain high-quality genome sequences for five representative isolates from each population (JP19, JP8, R22, JP44, and JP11), high-throughput genome sequencing was conducted using PacBio Sequel II and the circular consensus sequencing (CCS) mode at Novogene (Tianjin, China). It should be noted that the chromosomal-level assembly of R22 has already been published[7]. To achieve chromosomal-level genome assemblies for the remaining isolates, we constructed an SMRTbell library and employed PacBio Sequel II sequencing with high-fidelity (HiFi) reads in CCS mode. PacBio HiFi contigs were assembled using hifiasm v0.15.4-r342 with default parameter settings[60].

Subsequently, we employed a Hi-C scaffolding strategy to further organize and orient the draft scaffolds on chromosomes. To prepare the Hi-C library, fresh mycelia of *L. rhizohalophila* were harvested and subjected to formaldehyde fixation, crosslinking, endonuclease digestion, fill ends, biotin labeling, cyclization, purification, and DNA shearing. Paired-end sequencing using high-throughput sequencing technology was used to capture and sequence labeled target DNA fragments for the Hi-C library. The obtained clean data were subjected to quality control and filtering before alignment to the contigs assembled by hifiasm. This alignment process aimed to identify valid interaction pairs in the Hi-C data. For the genome anchoring of chromosomes, we employed 3D-DNA[61] to assist in clustering, sorting, and orienting the genome sequence. Manual adjustments were made using Juicerbox[62], including interrupting misjoined contigs, repositioning contigs, and correcting chromosome clusters, resulting in the final chromosome assembly. To annotate the protein-coding genes, a combination of homology-based and ab initio predictions was employed. Additionally, ~2 Gb of filtered RNA-seq reads was generated from fungal cultures in liquid medium to aid in gene annotation (details of RNA-seq analysis below). MAKER v2.31.9[63] was utilized to integrate gene sets predicted by various methods into a non-redundant and comprehensive gene set. The final reliable gene set was manually curated, and functional annotation was performed using Swiss-Prot[64], TrEMBL and KEGG[65], InterPro[66], and Gene Ontology (GO). To assess genome completeness, BUSCO v5.beta.1 was employed using with the OrthoDB fungi v10 (fungi_odb10) database. The search for repeat sequences, including tandem repeats and transposable elements (TEs), spanned the entire genome. Tandem repeats were annotated using TRF v4.09[67], whereas TE annotations were identified using ab initio strategies and homolog-based. For ab initio predictions, RepeatModeler v2.0.2a, RepeatScout v1.0.6, Piler v1.0, and LTR_FINDER v1.07[68,69] were performed with default parameters. RepeatMasker v4.1.2 and the associated RepeatProteinMask were used for homologous comparisons by searching against Repbase v21.01[70]. Detailed annotations of repeats were conducted using the Dfam v3.0 database of repetitive DNA families (https://dfam.org/home).

## Genome size evaluation using flow cytometry

To verify the genome sizes of certain *L. rhizohalophila* isolates suspected to be diploids (see "Results"), flow cytometry was employed as a complementary method to in silico genome assembly, providing additional insights into ploidy changes. To establish internal references, the genome sizes of *Arabidopsis thaliana* (2C = 133 Mb)[71] and *Fusarium pseudograminearum* Class2-1C (1C = 42.90 Mb)[72] were utilized. Prior to flow cytometry analysis, we prepared protoplasts from the five isolates. The fungus was inoculated into 100 ml of fresh potato dextrose broth (PDB) liquid medium in a 250 ml flask and shaken at 180 rpm at 20°C for 2 days. Mycelia were collected, washed twice with 0.6 M KCl (pH5.8) and lysed with a combination of 1% lysing enzyme (Sigma-Aldrich, USA), 2% cellulase (Sangon Biotech, China), and 2% snailase (Sangon Biotech, China). A protoplast suspension ($3-5 \times 10^6$ ml⁻¹) was prepared. Flow cytometry analysis was conducted as follows: 1 ml protoplast suspension was subjected to centrifugation for 10 min at $14,000 \times g$. After removing the supernatant, protoplasts were fixed in a mixture of methanol: acetic acid (3:1, v/v), 10% dimethyl sulfoxide (v/w), and 0.1% Triton X/100 (v/v) for 1 h at 4°C. Subsequently, the samples were chopped using a razor blade in Tris/MgCl$_2$ buffer (pH7.5) with 0.1 mg ml⁻¹ RNase A (Sangon Biotech, China). The suspension containing released nuclei was filtered through a 20 μm nylon filter to eliminate large debris and then incubated at 37°C for 15 min. Nuclei were stained with 50 μg ml⁻¹ propidium iodide (PI) (Fluka, Glossop, England) and immediately analyzed using an LSRII FACS machine (Becton Dickinson, NJ, USA). Each measurement consisted of 10,000 nuclei, and each isolate was measured three times. Flow cytometry data were

analyzed using FCS Express v7.0 (De Novo Software, Los Angeles, CA, USA)[73].

## Methods for identifying diploid-like genomic structures

To validate the potential of diploid-like hybrid formation, three approaches were used. First, Jellyfish v2.1.4, was employed to generate a k-mer spectrum for each sample using corrected and filtered short reads with a k-mer length of 17. Ploidy analysis was performed by comparing the sum of the k-mer pair coverage using the Smudgeplot v0.2.5 tool implemented in GenomeScope v2.0[74]. In addition, we employed the nucleus-specific stain 4′, 6-diamidino-2-phénylindole (DAPI, 5 µg ml⁻¹, Sigma-Aldrich) to determine the number of nuclei per hyphal cell. The isolates were inoculated on PDA with sterile coverslips inserted at a 45° angle and cultured until half of the coverslips reached the mycelial edge. Nuclei and cell walls were stained with DAPI and calcofluor white, respectively, and observed under a fluorescence microscope (Carl Zeiss AG, LSM900, Oberkochen, Germany). To calculate the nuclear and cell sizes, up to 600 or 100 hyphal cells were counted. Finally, the ratio of reads supporting alternative alleles to total reads in the VCF file was calculated for each heterozygous SNP to obtain the allele frequency value for each SNP. In theory, haploid isolates exhibit SNPs with allele frequencies of 1.0, whereas diploid isolates exhibit SNPs with allele frequencies of 0.5[75].

## Subgenome assignment

To facilitate the comparison of genomes and determine the parental origin, the JP19 and JP8 chromosomes were then accurately assigned to a given subgenome through a comparison with a haploid assembly using MUMmer v4.0.0[76]. The chromosome coverages of JP19 and JP8 were statistically compared, and JP19 and JP8 were divided into subgenomes A and B based on the difference in coverage[77].

## Macro-synteny and genomic variations

Syntenic gene pairs between the haploid isolates and the A and B subgenomes of the two diploids were identified using the MCScan pipeline implemented in JCVI v0.5.7 (https://zenodo.org/record/31631#.XpkUyTOeask)[78]. Gene collinearity was visualized using default parameters. In addition, small polymorphisms (SNPs and Indels) between haplotype genomes were extracted from whole-genome alignments using MUMmer v4.0.0. Pairwise identity between the seven haplotypes was calculated by dividing the respective query sequences into non-overlapping windows of 500 bp.

## Identification of the parental origin of two diploid hybrids

To determine the evolutionary origins of the diploid hybrids, we identified their potential haploid parents[16]. First, OrthoFinder v2.3.11 was used with default parameters to identify single-copy orthologous genes in the subgenomes of both the diploid and haploid genomes. An alignment matrix was generated by aligning all 7085 genes using MUSCLE v3.8.31, and a phylogenetic tree was constructed using RAxML v8.2.4 with 1000 bootstrap replicates. Heterozygous SNPs were observed during SNP calling in JP19 and JP8. To reduce false-positive calls, we used the FACET toolkit (https://github.com/aast242/FACET) to remove redundancy in BLAST searches for repetitive elements. The locations of all 13,404 genes were converted into BED files, and BCFtools v1.10.2 was used to filter the original VCF files and extract the intersection regions. We then calculated the number of heterozygous SNPs that matched all possible pairwise haploid combinations. Heterozygous SNPs were validated using PCR amplification and Sanger sequencing. Three representative contigs with dense heterozygous SNPs were selected and primer pairs were designed using Primer BioXM v2.6 (Table S6). PCR was performed under specific conditions, including denaturation, annealing, and extension. The PCR reaction volume was 50 µl, consisting of 25 µl 2×UniqueTM Taq Mix (Novogene, Beijing, China), 0.5 µl of each primer pair (50 µM), and 22 µl ultrapure H₂O.

## In vitro growth assays under a suite of abiotic stresses

To investigate whether diploids of *L. rhizohalophila* exhibit enhanced growth vigor compared with their progenitors in the context of saline adaptation, we conducted a series of abiotic stress sensitivity tests. The growth rates of the five isolates were determined in vitro[79] using different concentrations of NaCl (0, 0.2, 0.5, and 1 M) in PDA to simulate salinity stress conditions. We also examined the impact of pH ranging from 5 to 11 as a proxy for alkalinity stress conditions. The isolates were subjected to oxidative stress by adding two concentrations of hydrogen peroxide (1 mM and 4 mM) and menadione sodium bisulfite (0.05 mM and 0.2 mM) to PDA. Osmotic stress was assessed using sorbitol (0.2 M and 1.0 M) and KCl (0.2 M and 0.8 M). Furthermore, we evaluated the tolerance of fungal cell walls to Congo red (100 µg ml⁻¹ and 500 µg ml⁻¹) and calcofluor white (50 µg ml⁻¹ and 200 µg ml⁻¹). Four biological replicates were established, and all isolates were cultured in the dark at 25°C for 2 weeks to measure radial growth. Plugs were taken from the margins of freshly grown colonies for each replicate.

## Expression patterns in the two diploid hybrids

To investigate whether diploid formation was associated with transcriptional changes, we analyzed the expression profiles of these two diploids and three haploids using RNA-seq data. Four fungal growth conditions were established, including three gradients of salt stress (0 M NaCl, 0.3 M NaCl and 0.8 M NaCl) in vitro and a symbiotic status *in planta*, which are known to induce diverse phenotypes and variations in gene expression[9]. For the in vitro assay, fungal colonies were grown on PDA medium covered with cellophane membranes for 2 weeks at 24°C. The mycelium grown on each plate was gently scraped from the cellophane, collected, snap frozen in liquid nitrogen, and stored at −80°C until further analysis. For the *in planta* assay, fungal transcripts were investigated in infected roots. After 3 weeks of co-cultivation, the colonized roots were used for RNA extraction. Three independent biological replicates were used for each growth condition, resulting in 60 libraries. Raw sequencing reads were trimmed using fastp[80], and mapped to the JP11 reference genome using Hisat2[81]. Uniquely mapped and multimapped reads were assigned and counted using a custom pipeline that integrated featureCounts[82], mmquant[83], and custom Python and R scripts. Raw read counts were normalized using the TMM normalization approach to obtain Counts Per Million reads (CPMs), and further normalized by gene CDS lengths to obtain Fragments Per Kilobase of exon per million reads (FPKM) values using DESeq2[84] and edgeR[85]. Hierarchical clustering and principal component analysis were performed to explore the sample clustering patterns. We assessed the degree of subgenome dominance, also known as homoeolog expression bias[77]. Differentially expressed syntenic gene pairs with a fold change greater than two-fold were defined as dominant gene pairs. The statistical significance between biased and unbiased gene sets from diploids was evaluated using the t-test in R v3.6.3[86], applied to all gene models.

Two synthetic genomes (JP11 × R22 and JP11 × JP44) were created by concatenating genome sequences and annotations of their haploid ancestors (i.e., JP11, and R22, JP11, and JP44). These virtual diploid ancestors were compared to the actual hybrids JP19 and JP8 to identify syntenic gene pairs between JP11 × R22 and JP19, as well as between JP11 × JP44 and JP8. Differential expression analysis was then performed to identify genes with significant changes in expression before and after diploid formation, that is, between synthetic diploid ancestors and actual hybrids. Syntenic genes between JP11 × R22 and JP19, as well as JP11 × JP44 and JP8, were identified using a recently developed toolkit called Whole-Genome Duplication Integrated analysis (WGDI)[87]. We compared the expression of single-copy haploid genes with that of

their orthologs present in two homologous copies in diploids. Only genes with one copy in haploids and two homeologs in JP19 and JP8 were included in the differential expression analysis.

We next characterized allele expression patterns in the hybrids as described[88]. In brief, all hybrid reads were mapped to the haploid JP11 reference genome. In each diploid hybrid, allelic expression differences (additive and nonadditive) were determined by counting RNA-seq reads carrying at least two distinguishing SNP(s) between the two parents. Only genes with at least 20 allele-specific reads and less than 10% conflicting reads were used to determine the ratio of allelic expression in the hybrid. The additive model suggests that hybrid subgenomes exhibit gene expression levels similar to those of their parents. In this study, we focused on nonadditive and TUR genes, which display higher expression levels in hybrids than in both parents and are often associated with growth vigor (heterosis). To further explore the TUR genes, GO term enrichment analysis was performed using clusterProfiler v3.4.4[89]. We also compared TUR expression levels across all five isolates and growth conditions and visualized heat maps using the R v3.6.3 function heatmap (ggplot2).

### Staining and physiological approaches for evaluating the integrity of membrane lipids

We employed three approaches to evaluate the role of membrane lipids in the response to salt stress. First, we used the fluorescent dye, PI, at a concentration of $50\,\mu g\,ml^{-1}$ to assess membrane integrity. Five isolates were grown in PDB for 1 week, and the mycelia were stained with PI. Next, we visualized mycelial lipid droplets, which protect against membrane lipid peroxidation under various stressors[22], using Nile red at a concentration of $2.5\,\mu g\,ml^{-1}$. The size of the lipid droplets was measured using ImageJ software (National Institutes of Health, Bethesda, MD, USA). Thirdly, we estimated salt-induced membrane lipid peroxidation by measuring the content of MDA under 0 M, 0.3 M, and 0.8 M NaCl conditions[90].

### Statistical analysis

All statistical analyses were performed using the R v3.6.3. For the in vitro abiotic stress assays, semi-quantification of lipid droplets, and MDA measurement in mycelia, one-way analysis of variance (ANOVA) of the "stats" package was used for data analysis. Significant differences between means were determined using Duncan's multiple range test and Fisher's least significant difference (LSD) test function of the "agricolae" package at a significance level of $p < 0.05$. In cases where the data on nuclear size, protoplast diameter, cell width, and chromosome TE proportion did not meet the assumptions of normality and homogeneity of variance even after square root and log transformation, the non-parametric Kruskal–Wallis test was employed. Multiple comparisons were conducted using Conover's test with the post-hoc.kruskal.conover.test function of the PMCMR package. Additionally, Student's two-sample $t$-tests of the "stats" package were employed to assess differences in heterozygosity ratios between groups 4 and 5, as well as gene expression levels between subgenomes A and B from JP19 and JP8, at a significance level of $p < 0.05$.

### Reporting summary

Further information on research design is available in the Nature Portfolio Reporting Summary linked to this article.

## Data availability

The Illumina sequencing data for the 18 isolates have been deposited in the Sequence Read Archive (SRA) at NCBI under the accession numbers SRR23813528-SRR23813545. PacBio HiFi reads for JP19, JP8, JP44, and JP11 have been deposited in SRA under accession numbers SRR23908958–SRR23908961. The Hi-C reads from the five representative isolates have been deposited in SRA under the accession numbers SRR23901851–SRR23901855. Furthermore, the RNA-seq reads from the 60 libraries have been deposited in SRA under accession numbers SRR23343132–SRR23343191. The assembled genomes have been deposited in whole genome shotgun (WGS) under accession numbers JAULRH000000000, JAULRI000000000, JAULRJ0000 00000, JAUOZT000000000, and JAUPCS000000000. The sequencing data of additional 114 isolates have been deposited in the SRA under accession numbers SRR26626062-SRR26626175. All data have been submitted under the BioProject accession numbers PRJNA517533 and PRJNA931727. Annotations of genomes from the five isolates are provided in the Supplementary Materials. Source data are provided with this paper.

## Code availability

The source code implementing the analyses in this manuscript is available on Github (https://github.com/TaoFu123/FungiAnalysis.git) and Gitee (https://gitee.com/orionzhou/rnaseq/).

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

## Acknowledgements

This study was financially supported by the Fundamental Research Funds for the Central Non-profit Research of the Chinese Academy of Forestry (CAFYBB2020ZY002-1) and the National Natural Science Foundation of China (No. 32200097). This research was also supported by the Laboratory of Excellence ARBRE (ANR-11-LABX-0002-01) and the Huazhong Agricultural University, Wuhan, China (to F.M.M.).

## Author contributions

Z.Li and Z.Zhu conducted most experiments, analyzed the data, submitted the genomic and transcriptomic data, and wrote the manuscript. K.Qian and P.Zhou analyzed the transcriptomic data. B.Tang coordinated the field sampling and designed the experiment. B.Han and T.Fu performed the genome assembly using Hi-C scaffolding and part of bioinformatic analyses. Z.Zhong and E.Stukenbrock discussed the results. F.Martin analyzed the data, provided critical suggestions and edited the manuscript. Z.Yuan conceived, designed this study and wrote the manuscript. All authors edited and approved the manuscript.

## Competing interests

The authors declare no competing interests.

## Additional information

[1]State Key Laboratory of Tree Genetics and Breeding, Chinese Academy of Forestry, 100091 Beijing, China. [2]Research Institute of Subtropical Forestry, Chinese Academy of Forestry, Hangzhou 311400, China. [3]Nanjing Forestry University, Nanjing 100071, China. [4]College of Life Science, Zhejiang University, Hangzhou 310058 Zhejiang, China. [5]Department of Animal, Plant and Soil Science, School of Agriculture, Biomedical and Environmental Sciences, La Trobe University, Bundoora, VIC 3086, Australia. [6]Jiangsu Key Laboratory for Bioresources of Saline Soils, School of Wetlands, Yancheng Teachers University, Yancheng 224002, China. [7]State Key Laboratory of Systematic and Evolutionary Botany, Institute of Botany, Chinese Academy of Sciences, 100093 Beijing, China. [8]Ministry of Education Key Laboratory for Bio-Resource and Eco-Environment, College of Life Sciences, Sichuan University, Chengdu 610065, China. [9]Shenzhen Zhuoyun Haizhi Medical Research Center Co., Ltd, Shenzhen 518063, China. [10]National Key Facility for Crop Gene Resources and Genetic Improvement, Institute of Crop Sciences, Chinese Academy of Agricultural Sciences, 100081 Beijing, China. [11]Environmental Genomics, Christian-Albrechts University, 24118 Kiel, Germany. [12]Max Planck Fellow Group Environmental Genomics, Max Planck Institute for Evolutionary Biology, 24306 Plön, Germany. [13]Université de Lorraine, INRAE, UMR Interactions Arbres/Microorganismes, Centre INRAE Grand Est—Nancy, 54280 Champenoux, France. [14]These authors contributed equally: Zhongfeng Li, Zhiyong Zhu, Kun Qian, Boping Tang. ✉e-mail: pzhou@caas.cn; francis.martin@inrae.fr; yuanzl@caf.ac.cn

