## [Peer Review File · Nature Communications]

Intraspecific diploidization of a halophyte root fungus drives heterosisEditorial Note: Parts of this Peer Review File have been redacted as indicated to remove third-party material where no permission to publish could be obtained.

REVIEWER COMMENTS

Reviewer #1 (Remarks to the Author):

In this paper, Li and colleagues examined evolution of genomes of *Laburnicola rhizohalophila*, a root dark-septate endophytic fungus sampled in a salt marsh. They found that JP19 and JP8 exhibit nearly doubled genome sizes, gene numbers, and chromosome numbers, which were attributed to the intraspecific hybridization. In addition, they examined the differences in varying numbers of aspects (e.g., expression level, SNP counts, and abiotic stressors) between JP19 and other strains. Lastly, they narrowed down two important pathways and validated them with some experiments.

Overall, the work is well written, their findings are novel, and the evidences are sufficient to explain for the diploid hybrid endophyte. Here I have some concerns (most of them are minor) that authors might wish to address.

- 1) I recommend expanding the introduction to provide more context and explain the significance of intraspecific hybridization in the field. This will help readers better understand the motivation behind using the root dark-septate endophytic fungus for this study.
- 2) Line 190, please double-check the space before "angle"
- 3) In the M&M, please include versions of software and packages R. Some have included them, but some do not.
- 4) Please specify your background or criteria when conducting GO enrichment.
- 5) Lines 331-335. This sounds contradictory. Please rephrase them.
- 6) Lines 370-376, Please provide information on the identity of duplicated genes and whether there are variations between them. Consider exploring if one copy differs from another in terms of RNA expression.
- 7) Lines 405-406, you identified unique and overlapping gene families and then you fully stopped. Please provide a brief commentary on their significance or characteristics.
- 8) Based on these observations, JP19 and JP8 can be considered intraspecific diploid hybrids. This is too arbitrary. consider using "could" or something similar instead of "can" to acknowledge the possibility without making a conclusive statement without wet data.
- 9) Lines 310-313, please ensure that your annotations are publicly available for other researchers to access and reproduce your results. Consider providing details on where these annotations can be accessed.

Reviewer #2 (Remarks to the Author):

Li et al. delved into the genomic evolution of the dark-septate endophytic fungus *Laburnicola rhizohalophila*, isolated from roots of the seepweed *Suaeda salsa*, thriving within salt marsh environments. Through the generation of chromosome-level assemblies from five isolates, they uncovered significant genome plasticity attributed to chromosomal polymorphisms. Analyses provided compelling evidence supporting the hybrid origin of intraspecific diploid-like genomic structures. Of particular interest was the observation of one specific diploid phenotype, denoted as JP19, which showed enhanced growth under abiotic stress conditions, such as exposure to 0.3 M NaCl, potentially due to the upregulation of genes related to the synthesis of glycerolipid membranes. This study sheds new light on the phenomenon of diploidization in fungi and its pivotal role in evolutionary adaptation to salinity stress conditions. Furthermore, the authors introduce a new model aimed at exploring the implications of diploidization in the evolution of fungi that form mutualistic associations with roots, which could signify the initial stride toward the development of enhanced microbial inoculants.

The manuscript is very well-written, and the quality of the figures is good. The methods are clear and thoroughly described. The results were carefully interpreted, and the conclusions are well supported. The work seems very innovative to me. In my opinion, the article is suitable for publication in *Nature Communications*, after addressing the following minor issues:

Main Text:

Line 92: Although it is evident from the reference provided (Yuan et al., 2021) and Fig. 1B, I would recommend explicitly stating in the main text that fungi were isolated from Suaeda salsa.

Line 189: "diamidino-2-phénylindole" should be corrected to "4',6-diamidino-2-phenylindole"

Line 202: " ... using MUMmer v4.0.0".

Line 341: "half" should be corrected to "twice"

Lines 374-375: Could you please provide the number/percentage of duplicated BUSCO?

Line 407: Please place the reference to Fig. 4A (phylogenetic tree) in the first paragraph of the results section entitled "Pinpointing the diploid formation via parasexual hybridization".

Line 585: "Comapre" should be corrected to "Compared"

Throughout the manuscript: ensure consistent usage of terms such as "subgenome" or "sub-genome", "subgenomic" or "sub-genomic", etc.

Throughout the manuscript: Please check missing references, such as Kurtz et al. (2004) for MUMmer v4.0.0. However, in this case, since you are using MUMmer v4.0.0, it would be better to cite: Marçais G, Delcher AL, Phillippy AM, Coston R, Salzberg SL, Zimin A. MUMmer4: A fast and versatile genome alignment system. PLoS computational biology. 2018 Jan 26;14(1):e1005944.

Supplementary Materials:

Line 22: "peaks" should be corrected to "peak"

Line 54: "exposure" should be corrected to "exposed"

Line 57: "one way" should be corrected to "one-way"

Line 73: "exposure" should be corrected to "exposed"

Line 76: "one way" should be corrected to "one-way"

Reviewer #3 (Remarks to the Author):

In this study, the authors performed a population genomics study on 47 strains of a root dark-septate endophytic fungus *Laburnicola rhizophila* from a salt marsh and identified diploid strains originated from hybridization events between haploid strains of the species. They found that one diploid strain exhibited growth vigor under certain stressful conditions. They then performed comparative RNA-seq analysis and found that a set of genes involved in the synthesis of glycerolipid membranes were upregulated in the diploid strain grown in 0.3M NaCl compared to its assumed haploid parents, leading to the conclusion that the upregulation of these genes contributes to the growth vigor or heterosis of the diploid strain.

The study is interesting and provides new insights into the divergence and adaptive evolution of asexual endophytic fungi. The diploid state and the origin of the genome duplication of the two strains are supported by sufficient solid evidence. However, several important issues need to be addressed or clarified further.

General comments

1. The origins and related information of the 47 strains of *L. rhizohalophila* employed in this study should be given in a supplementary table and strains in each of the five groups should be specified. This will clarify whether the two diploid strains JP8 and JP19 which were extensively studied and the assumed parents of strain JP19 belong to the same or different groups. It is important to know how many strains are there in the groups containing the diploid strains JP8 and JP19 and whether these strains are all diploid.
2. For an evolution and population-level study, multiple strains are usually required to represent a lineage. At least for the growth assay, all 47 strains should be employed to confirm the growth vigor of strain JP19 and to show if the other strains in the same group exhibit similar growth vigor.
3. If more diploid strains can be found to exhibit similar phenotypes with strain JP19, these strains should be employed in detailed comparative genomic and transcriptomic studies to enhance the conclusion of this study. If only very limited diploid strains showing growth vigor have been found, then the authors need to explain why these strains with elevated fitness as claimed in this study have not expanded their populations.
4. In the RNA-seq analysis, genes showing down-regulation in the diploid strain are probably also important to explain the growth vigor. Downregulation of certain genes in heterotic strains probably indicates improved homeostasis of the related metabolomic pathways in the diploid strains compared to their haploid parents under stressful conditions. Thus, the functions of the TDR genes are probably worthy of further analysis.
5. The molecular data supporting the proposed mechanisms underlying the growth vigor of the diploid strain as shown in the working model (Fig. 8) should be clearly and specifically presented and discussed in the Results and Discussion sections. 'Ion transmembrane transporter' is highlighted in the model, but the genes concerned and their regulation level are not specified in the Result, though are shown in Fig. 7B. The genes involved in phospholipid biosynthesis are shown in the model, but they are not mentioned in the Result and no data is given to show if these genes are upregulated.

Specific comments

- L.95, a reference should be given for the isolation and identification methods of the new strains.
- L.109, the sequence coverages should be given.
- L.116, how the heterozygous SNPs were treated in the phylogenetic analysis?
- L.170, the method of protoplast generation should be given.
- L.246, the method for the creation of synthetic genomes should be given. How and where were these synthetic genomes used?
- L.253, why only two diploid strains were subjected to RNA-seq? If haploid strains were not employed in RNA-seq, how were the gene expression levels in the diploid and haploid strains compared?
- L.324, how about the heterozygosity of the other three groups? Even a haploid genome will have heterozygous sites.
- L.329, sectors are usually formed by heterokaryon strains. How does a diploid strain form sectors?
- L.334, why were strains from Groups 4 and 5 not included in the het gene comparison? I doubt the relevance of het genes to this study since heterokaryosis was not found in the strains employed. Fig. 1G is probably not necessary.

L.340, I suggest including strains from Groups 1-3 in Fig. 2A & B for direct comparison.

L.357, I cannot find any difference between strains JP8 and JP44 in Fig. 2E.

L.375, the discussion "the result of mixed infections with their closest extant relatives" is not relevant to the data presented.

Ls.412-414: The sentences here are not logical, for only one clade is mentioned and the clade mentioned does not contain JP19B.

Ls.465-479, sub-genome dominance expression: It is difficult for me to understand the data shown in Fig. 6B. What values were compared to get the p values? For example, for strain JP19 under the condition 0M NaCl, $N = 888$ for A is almost the same as $N = 889$ for B, why a $p = 0.002$ was obtained?

L.490 and Fig. 6C, what is the category for the bottom block in blue?

L.501, the difference in additive and nonadditive gene expression patterns between JP19 and JP8 should be described here.

L.521, Fig. 7C does not show PI staining.

L.543. It should be cautious to say "Notably, the chromosome numbers of haploid *L. rhizohalophila* surpassed those of the most ascomycetous lineages". Please check if most ascomycetous lineages have less than 21 chromosomes.

L549, I doubt the relevance of the discussion "Frequent heterokaryosis fuels promiscuous and recurrent hybridizations" with this study, since heterokaryon was not observed in the fungus tested.

Point-by-point responses to the reviewers' comments

Reviewer #1 (Remarks to the author)

In this paper, Li and colleagues examined evolution of genomes of *Laburnicola rhizohalophila*, a root dark-septate endophytic fungus sampled in a salt marsh. They found that JP19 and JP8 exhibit nearly doubled genome sizes, gene numbers, and chromosome numbers, which were attributed to the intraspecific hybridization. In addition, they examined the differences in varying numbers of aspects (e.g., expression level, SNP counts, and abiotic stressors) between JP19 and other strains. Lastly, they narrowed down two important pathways and validated them with some experiments.

Action. We appreciate the reviewers' positive comments and valuable suggestions regarding our manuscript.

Overall, the work is well written, their findings are novel, and the evidences are sufficient to explain for the diploid hybrid endophyte. Here I have some concerns (most of them are minor) that authors might wish to address.

1) I recommend expanding the introduction to provide more context and explain the significance of intraspecific hybridization in the field. This will help readers better understand the motivation behind using the root dark-septate endophytic fungus for this study.

Action. We have expanded the Introduction section as suggested by introducing the context and significance of interspecific and intraspecific fungal hybridization in lines 86-88 and made minor revisions in the following paragraph (lines 88-94).

2) Line 190, please double-check the space before "angle"

Action. We checked the space and used the correct form of "45°".

3) In M&M, please include versions of the software and packages R. Some have included them, but some do not.

Action. We have provided all versions of the software and packages in R in the revised version.

4) Please specify your background or criteria when conducting GO enrichment.

Action. GO term enrichment analysis was performed using all 864 genes with differential upregulation when the mycelium was grown on 0.3 M NaCl. Nineteen GO terms were identified (see table below). In the main text, Fig. 7A shows only the top 15 GO terms according to the rank of the *p* adjusted values.

ID	Description	p .adjust	Count
GO:0046470	phosphatidylcholine metabolic process	0.001766785	9

GO:0006629	lipid metabolic process	0.002012483	53
GO:0046486	glycerolipid metabolic process	0.007065408	22
GO:0010876	lipid localization	7.07E-03	14
GO:0044255	cellular lipid metabolic process	0.007065408	43
GO:0006658	phosphatidylserine metabolic process	7.07E-03	6
GO:0015850	organic hydroxy compound transport	0.007065408	9
GO:0006656	phosphatidylcholine biosynthetic process	7.07E-03	7
GO:0006650	glycerophospholipid metabolic process	0.011606052	19
GO:0097164	ammonium ion metabolic process	0.018976484	9
GO:0016042	lipid catabolic process	0.020982809	16
GO:0006869	lipid transport	0.022210787	12
GO:0051403	stress-activated MAPK cascade	0.022210787	7
GO:0071214	cellular response to abiotic stimulus	0.022210787	18
GO:0104004	cellular response to environmental stimulus	0.022210787	18
GO:0006644	phospholipid metabolic process	0.023769737	22
GO:0015980	energy derivation by oxidation of organic compounds	0.041563027	18
GO:0015849	organic acid transport	0.041563027	16
GO:0006812	cation transport	0.04833329	25

5) Lines 331-335. This sounds contradictory. Please rephrase them.

Action. The notion that frequent parasexual hybridization is due to a high number of het genes is not entirely supported by our data. Consequently, in the revision, we omitted this hypothesis to prevent confusion while maintaining the integrity of our main conclusions.

6) Lines 370-376, Please provide information on the identity of duplicated genes and whether there are variations between them. Consider exploring if one copy differs from another in terms of RNA expression.

Action. We have provided detailed information on the identities of the duplicated genes identified by BUSCO (Table 1). More specifically, JP19 contained 10,731 duplicated genes and JP8 contained 10,662 duplicated genes. To test whether one copy differs from another in terms of RNA expression, we assessed the degree of subgenome dominance, also known as homoeolog expression bias. The majority of homoeologous gene pairs (67.4-70.9%) exhibited unbiased expression. Specifically, homoeologs in JP19 displayed higher expression in subgenome A than in subgenome B under all the tested conditions. The detailed results are provided in Fig. 6B and the main text (lines 490-503).

7) Lines 405-406, you identified unique and overlapping gene families and then you fully stopped. Please provide a brief commentary on their significance or characteristics.

Action. We have added a brief commentary on the unique and overlapping gene families, as well as their potential evolutionary significances in lines 426-429.

8) Based on these observations, JP19 and JP8 can be considered intraspecific diploid hybrids. This is too arbitrary. consider using "could" or something similar instead of "can" to acknowledge the possibility without making a conclusive statement without wet data.

Action. We revised the manuscript accordingly.

9) Lines 310-313, please ensure that your annotations are publicly available for other researchers to access and reproduce your results. Consider providing details on where these annotations can be accessed.

Action. Apologies for oversight. The structural and functional annotations of five genomes are now included in the Supplementary Materials section.

Reviewer #2 (Remark to the author)

Li et al. delved into the genomic evolution of the dark-septate endophytic fungus *Laburnicola rhizohalophila*, isolated from roots of the seepweed *Suaeda salsa*, thriving within salt marsh environments. Through the generation of chromosome-level assemblies from five isolates, they uncovered significant genome plasticity attributed to chromosomal polymorphisms. Analyses provided compelling evidence supporting the hybrid origin of intraspecific diploid-like genomic structures. Of particular interest was the observation of one specific diploid phenotype, denoted as JP19, which showed enhanced growth under abiotic stress conditions, such as exposure to 0.3 M NaCl, potentially due to the upregulation of genes related to the synthesis of glycerolipid membranes. This study sheds new light on the phenomenon of diploidization in fungi and its pivotal role in evolutionary adaptation to salinity stress conditions. Furthermore, the authors introduce a new model aimed at exploring the implications of diploidization in the evolution of fungi that form mutualistic associations with roots, which could signify the initial stride toward the development of enhanced microbial inoculants.

The manuscript is very well-written, and the quality of the figures is good. The methods are clear and thoroughly described. The results were carefully interpreted, and the conclusions are well supported. The work seems very innovative to me. In my opinion, the article is suitable for publication in Nature Communications, after addressing the following minor issues:

Action. We thank the reviewer for her/his positive assessment and valuable suggestions regarding our manuscript.

Line 92: Although it is evident from the reference provided (Yuan et al., 2021) and Fig. 1B, I would recommend explicitly stating in the main text that fungi were isolated from *Suaeda salsa*.

Action. Done

Line 189: "diamidino-2-phénylindole" should be corrected to "4',6-diamidino-2-phenylindole"

Action. Done.

Line 202: " ... using MUMmer v4.0.0".

Action. Done.

Line 341: "half" should be corrected to "twice"

Action. Done.

Lines 374-375: Could you please provide the number/percentage of duplicated BUSCO?

Action. The percentage of duplicated BUSCO is provided in Table 1.

Line 407: Please place the reference to Fig. 4A (phylogenetic tree) in the first paragraph of the results section entitled "Pinpointing the diploid formation via parasexual hybridization".

Action. Done. We have inserted Fig. 4A after “we utilized 7,085 single-copy genes to infer the phylogeny” in the first paragraph of the results section.

Line 585: "Comapre" should be corrected to "Compared"

Action. Done.

Throughout the manuscript: ensure consistent usage of terms such as "subgenome" or "sub-genome", "subgenomic" or "sub-genomic", etc.

Action. Done.

Throughout the manuscript: Please check missing references, such as Kurtz et al. (2004) for MUMmer v4.0.0. However, in this case, since you are using MUMmer v4.0.0, it would be better to cite: Marçais G, Delcher AL, Phillippy AM, Coston R, Salzberg SL, Zimin A. MUMmer4: A fast and versatile genome alignment system. PLoS computational Biology. 2018 Jan 26;14(1):e1005944.

Action. Done.

Supplementary material

Line 22: "peaks" should be corrected to "peak"

Action. Done.

Line 54: "exposure" should be corrected to "exposed"

Action. Done.

Line 57: "one way" should be corrected to "one-way"

Action. Done.

Line 73: "exposure" should be corrected to "exposed"

Action. Done.

Line 76: "one way" should be corrected to "one-way"

Action. Done.

Reviewer #3 (Remark to the author)

In this study, the authors performed a population genomics study on 47 strains of a root dark-septate endophytic fungus *Laburnicola rhizohalophila* from a salt marsh and identified diploid strains originated from hybridization events between haploid strains of the species. They found that one diploid strain exhibited growth vigor under certain stressful conditions. They then performed comparative RNA-seq analysis and found that a set of genes involved in the synthesis of glycerolipid membranes were upregulated in the diploid strain grown in 0.3M NaCl compared to its assumed haploid parents, leading to the conclusion that the upregulation of these genes contributes to the growth vigor or heterosis of the diploid strain.

The study is interesting and provides new insights into the divergence and adaptive evolution of asexual endophytic fungi. The diploid state and the origin of the genome duplication of the two strains are supported by sufficient solid evidence. However, several important issues need to be addressed or clarified further.

Action. We thank the reviewer for her/his constructive comments regarding our manuscript.

General comments

1. The origins and related information of the 47 strains of *L. rhizohalophila* employed in this study should be given in a supplementary table and strains in each of the five groups should be specified. This will clarify whether the two diploid strains JP8 and JP19 which were extensively studied and the assumed parents of strain JP19 belong to the same or different groups. It is important to know how many strains are there in the groups containing the diploid strains JP8 and JP19 and whether these strains are all diploid.

Action. We apologize for the oversight. We have added this information in Table S1.

2. For an evolution and population-level study, multiple strains are usually required to represent a lineage. At least for the growth assay, all 47 strains should be employed to confirm the growth vigor of strain JP19 and to show if the other strains in the same group exhibit similar growth vigor.

Action. Agreed. It is customary to use multiple strains to represent lineages. In fact, we have already performed a comparison between all isolates to substantiate the growth vigor of JP19. This comparison validated our original claim that JP19 demonstrated superior growth compared with all haploids. However, our primary goal was to compare JP19 with all its parent isolates, as illustrated in Fig. S6. Therefore, we maintain that current data presentation is more appropriate and effective.

To alleviate your concerns, we include the remaining haploid and diploid isolates for analyses (see below). Two main conclusions emerge. First, the growth vigor of JP19 is still apparent when compared to all tested haploid isolates under four abiotic stress conditions. Second, the other diploids do not show any sign of heterosis, further supporting our claims.

3. If more diploid strains can be found to exhibit similar phenotypes with strain JP19, these strains should be employed in detailed comparative genomic and transcriptomic studies to enhance the conclusion of this study. If only very limited diploid strains showing growth

vigor have been found, then the authors need to explain why these strains with elevated fitness as claimed in this study have not expanded their populations.

Action. We discovered two diploid lineages and analyzed their potential for heterosis (Figure 1). Surprisingly, only one of the two diploid lineages, JP19, exhibited growth vigor compared with its haploid parents. Therefore, we explored possible explanations for why another diploid lineage, JP8, did not show any indications of heterosis. Additionally, we observed poor growth vigor in the JP32 isolate, which clustered with JP19, during our *in vitro* growth assay on PDA plates. We suspect that the unstable diploid genome in JP32 may be responsible for this observation. Furthermore, we noticed that the other three isolates clustered with JP8 also contained sectors in their cultures. However, we did not observe any sectors associated with JP19 during our long-term laboratory experiments spanning from 2014 to the present. We believe that the frequency of sectoring is a useful indicator of the degree of growth vigor. Unfortunately, we have not sequenced the entire genome of JP32 using the PacBio-HiFi platform, which prevents us from providing more evidence on the genomic differences between JP19 and JP32.

4. In the RNA-seq analysis, genes showing downregulation in the diploid strain were probably also important for explaining growth vigor. Downregulation of certain genes in heterotic strains probably indicates improved homeostasis of the related metabolomic pathways in the diploid strains compared to their haploid parents under stressful conditions. Thus, the functions of the TDR genes are probably worthy of further analysis.

Action. As mentioned above, GO term enrichment analysis was performed for all 778 genes that showed differential downregulation. A total of 39 GO terms were identified (see table below). We did not find any GO terms directly related to salt-responsive pathways, while most GO terms were associated with the mitotic cell cycle, chromatin dynamics, and microtubule-related processes in promoting cell proliferation and maintaining the normal structure of chromosomes. Therefore, we have not discussed these pathways further in the Discussion section.

ID	Description	p.adjust	Count
GO:0006520	cellular amino acid metabolic process	0.000161583	41
GO:0090307	mitotic spindle assembly	0.002442787	10
GO:0051225	spindle assembly	0.002442787	10
GO:0006082	organic acid metabolic process	0.003309593	56
GO:0006418	tRNA aminoacylation for protein translation	0.003955552	12
GO:0043038	amino acid activation	0.0046397	12
GO:0043039	tRNA aminoacylation	0.0046397	12
GO:0043436	oxoacid metabolic process	0.004700557	54
GO:0019752	carboxylic acid metabolic process	0.005030253	53
GO:0000022	mitotic spindle elongation	0.007734122	8
GO:0051231	spindle elongation	0.007734122	8
GO:0009119	ribonucleoside metabolic process	0.008472799	9

GO:0007020	microtubule nucleation	0.008472799	7
GO:0002181	cytoplasmic translation	0.008472799	28
GO:0007051	spindle organization	0.009882174	12
GO:0007052	mitotic spindle organization	0.010721927	11
GO:0007080	mitotic metaphase plate congression	0.011215047	9
GO:1903047	mitotic cell cycle process	0.012336925	49
GO:0031109	microtubule polymerization or depolymerization	0.012381191	9
GO:0051310	metaphase plate congression	0.012381191	9
GO:0000226	microtubule cytoskeleton organization	0.01602432	17
GO:0008608	attachment of spindle microtubules to kinetochore	0.020531804	10
GO:0000278	mitotic cell cycle	0.020872853	49
GO:0007017	microtubule-based process	0.020872853	19
GO:0140014	mitotic nuclear division	0.022638867	27
GO:0072528	pyrimidine-containing compound biosynthetic process	0.024464909	9
GO:0046785	microtubule polymerization	0.025333806	8
GO:1901657	glycosyl compound metabolic process	0.027710041	11
GO:0046131	pyrimidine ribonucleoside metabolic process	0.03007456	5
GO:1901605	alpha-amino acid metabolic process	0.03007456	25
GO:0072527	pyrimidine-containing compound metabolic process	0.03007456	9
GO:0009116	nucleoside metabolic process	0.03007456	10
GO:0042278	purine nucleoside metabolic process	0.03007456	6
GO:1902850	microtubule cytoskeleton organization involved in mitosis	0.037011685	11
GO:0051303	establishment of chromosome localization	0.040781886	9
GO:0007010	cytoskeleton organization	0.046053435	29
GO:0098813	nuclear chromosome segregation	0.04915271	26
GO:0043094	cellular metabolic compound salvage	0.04915271	7
GO:0007059	chromosome segregation	0.04915271	27

5. The molecular data supporting the proposed mechanisms underlying the growth vigor of the diploid strain as shown in the working model (Fig. 8) should be clearly and specifically presented and discussed in the Results and Discussion sections. ‘Ion transmembrane transporter’ is highlighted in the model, but the genes concerned and their regulation level are not specified in the Result, though are shown in Fig. 7B. The genes involved in phospholipid biosynthesis are shown in the model, but they are not mentioned in the Result and no data is given to show if these genes are upregulated.

Action. The Results sections of the revised manuscript contain additional interpretations of the gene expression patterns of interest (lines 533-547). We previously emphasized that the genes responsible for phospholipid biosynthesis and ion transport were significantly upregulated in JP19 under 0.3 M NaCl.

Specific comments

L.95, a reference should be given for the isolation and identification methods of the new strains.

Action. Done.

L.109, the sequence coverages should be given.

Action. Done.

L.116, how the heterozygous SNPs were treated in the phylogenetic analysis?

Action. Based on our practical experience, we employed degenerate bases in phylogenetic analysis to handle heterozygous SNPs. This method is widely used and has been emphasized in the figure legend for clarity.

Degenerate bases symbol	Explanation	Nucleotides
R	puRine	A/G
Y	pYrimidine	C/T
M	aMino	A/C
K	Keto	G/T
S	Strong	G/C
W	Weak	A/T
B	not A (B comes after A)	G/T/C
V	not T (V comes after T and U)	G/A/C
D	not C (D comes after C)	G/A/T
H	not G (H comes after G)	A/C/T
N	any Nucleotide (not a gap)	A/T/C/G

L.170, the method of protoplast generation should be given.

Action. Done.

L.246, the method for the creation of synthetic genomes should be given. How and where were these synthetic genomes used?

Action. Done.

L.253, why only two diploid strains were subjected to RNA-seq? If haploid strains were not employed in RNA-seq, how were the gene expression levels in the diploid and haploid strains compared?

Action. Apologies for oversight. The description of the RNA-seq analysis of the diploid and haploid samples is missing in the original version of the Methods section. We have now included this information in the Methods and Materials section.

L.324, how about the heterozygosity of the other three groups? Even a haploid genome will have heterozygous sites.

Action. Agreed. Based on the allele frequency plot in the Fig. 2B, it appeared that some heterozygous sites were present in the haploids during SNP calling. It has been frequently addressed that the high numbers of false positive heterozygous SNPs can arise in repetitive or homologous sequence regions (Shen et al., 2010; Schrader et al., 2014; Ewing, 2015). The *L. rhizohalophila* genome is featured by its high TE content. Thus, it is not surprising that the heterozygous SNPs could be detected, though theoretically speaking no heterozygosity occur in haploid organisms. To reduce false-positive calls, we removed the TE regions in the reference genome for reads mapping and SNP calling. In this case, the number of heterozygous SNPs in three haploid groups was rather limited and the expected amount of heterozygosity is almost zero (see table below).

Haploid groups	Average heterozygosity (%)
Group 1	0.0047
Group 2	0.0132
Group 3	0.0296

Schrader, L., Kim, J. W., Ence, D., Zimin, A., Klein, A., Wyschetzki, K., Weichselgartner, T., Kemena, C., Stökl, J., Schultner, E., Wurm, Y., Smith, C. D., Yandell, M., Heinze, J., Gadau, J., & Oettler, J. (2014). Transposable element islands facilitate adaptation to novel environments in an invasive species. *Nature Communications*, 5, 5495.

Shen, Y., Wan, Z., Coarfa, C., Drabek, R., Chen, L., Ostrowski, E. A., Liu, Y., Weinstock, G. M., Wheeler, D. A., Gibbs, R. A., & Yu, F. (2010). A SNP discovery method to assess variant allele probability from next-generation resequencing data. *Genome Research*, 20(2), 273–280.

Ewing A. D. (2015). Transposable element detection from whole genome sequence data. *Mobile DNA*, 6, 24.

L.329, sectors are usually formed by heterokaryon strains. How does a diploid strain form sectors?

Action. In groups 4 and 5 in our study, we observed that only JP19 demonstrated stable growth during cultivation. In contrast, JP8 and other diploid strains frequently exhibited colony sectors (Fig. 1F). We hypothesized that this instability in diploid strains is due to chromosome loss, leading to the reversion to the haploid form. This phenomenon is supported by previous studies by Engel et al. (2020) and Banuett (2015), who reported similar findings in diploid strains treated with benomyl (a haploidizing agent). These studies found that the colonies grew irregularly, accompanied by typical sectoring of the colony, indicating that the sectors were haploid or aneuploid segregants from the original diploid colony.

Engel, T., Verweij, P. E., van den Heuvel, J., Wangmo, D., Zhang, J., Debets, A. J. M., & Snelders, E. (2020). Parasexual recombination enables *Aspergillus fumigatus* to persist in cystic fibrosis. *ERJ Open Research*, 6(4): 00020–2020.

Banuett F. (2015). From dikaryon to diploid. *Fungal Biology Reviews*. 29: 194–208.

L.334, why were strains from Groups 4 and 5 not included in the *het* gene comparison? I doubt the relevance of *het* genes to this study since heterokaryosis was not found in the strains employed. Fig. 1G is probably not necessary.

Action. Heterozygous (*het*) genes were compared between the three *L. rhizohalophila* haploids and other haploid fungal species, based on published data. The strains from groups 4 and 5 were diploid; thus, they were not considered in the comparison of *het* genes. We acknowledge your comment and agree that our previous contention regarding the frequent occurrence of parasexual hybridization due to *het* genes is not strongly supported by our data. Therefore, we have removed this section from the revised version of our work.

L.340, I suggest including strains from Groups 1-3 in Fig. 2A & B for direct comparison.

Action. Done.

L.357, I cannot find any difference between strains JP8 and JP44 in Fig. 2E.

Action. Agreed. We have not documented these results in our previous submission. Nevertheless, we conducted a comparison between diploid and three haploid isolates, as indicated by the asterisks in Fig. 2E. As a result, we have revised the sentence to "Significant differences were observed in terms of nuclear size ($\chi^2 = 558.36$, $P < 0.001$), protoplast size ($\chi^2 = 86.84$, $P < 0.001$), length ($F = 57.38$, $P < 0.001$), and hyphal cell width ($\chi^2 = 104.48$, $P < 0.001$) between the diploid and the three haploid isolates."

L.375, the discussion “the result of mixed infections with their closest extant relatives” is not relevant to the data presented.

Action. We have deleted this sentence accordingly.

LS.412-414: The sentences here are not logical, for only one clade is mentioned and the clade mentioned does not contain JP19B.

Action. Agreed. We have added a description of another monophyletic clade containing JP11, JP19^B, and JP8^B that clustered together.

LS.465-479, sub-genome dominance expression: It is difficult for me to understand the data shown in Fig. 6B. What values were compared to get the p values? For example, for strain JP19 under the condition 0 M NaCl, N = 888 for A is almost the same as N = 889 for B, why a p = 0.002 was obtained?

Action. There are two aspects to consider in the subgenome dominance expression analysis shown in Fig. 6B. Initially, the expression levels of the homologous gene pairs in the two subgenomes were examined. The N values represent the number of genes in subgenomes A and B that were dominantly expressed. Furthermore, *p*-values were calculated by conducting *t*-tests to compare the differences in expression levels between all homologous genes in subgenomes A and B.

L.490, and Fig. 6C, what is the category for the bottom block in blue?

Action. This category refers to unclassified genes. We have added this information to Fig. 6C.

L.501, the difference in additive and non-additive gene expression patterns between JP19 and JP8 is described here.

Action. We have added this information to the Results section.

L. 521 Fig. 7C did not show PI staining.

Action. The data are provided in Supplementary Fig. S9.

L.543. It should be cautious to say “Notably, the chromosome numbers of haploid *L. rhizohalophila* surpassed those of the most ascomycetous lineages”. Please check if most ascomycetous lineages have less than 21 chromosomes.

Action. Wang et al. (2020) and Pfeffer et al. (2023) have conducted extensive researches on the range of chromosome numbers in 21 orders of Ascomycota, which generally have 3-8 chromosomes. The sole exception is the orders Dothideales and Capnodiales, where some species possess 21 chromosomes, which is the highest number reported among haploid ascomycetes. Our analysis suggests that the chromosome numbers of haploid *L. rhizohalophila* are higher than those of the most ascomycetous lineages.

Wang, B., Liang, X., Gleason, M. L., Hsiang, T., Zhang, R., & Sun, G. (2020). A chromosome-scale assembly of the smallest Dothideomycete genome reveals a unique genome compaction mechanism in filamentous fungi. *BMC genomics*, 21(1), 321.

Pfeffer, B., Lymbery, C., Booth, B., & Allen, J.L. (2023) Chromosomal genome sequence assembly and mating-type (*MAT*) locus characterization of the leprose asexual lichenized fungus *Lepraria neglecta* (Nyl.) Erichsen. *Lichenologist*, 55, 41–50.

[REDACTED]

L549, I doubt the relevance of the discussion “Frequent heterokaryosis fuels promiscuous and recurrent hybridizations” with this study, since heterokaryon was not observed in the fungus tested.

Response: Thank you for your helpful comments. We revised the description in this section.

REVIEWERS' COMMENTS

Reviewer #1 (Remarks to the Author):

The authors have adequately addressed my concerns. I don't have any further questions.

Reviewer #3 (Remarks to the Author):

I appreciate the efforts of the authors to accommodate the majority of my concerns in the revised version. Regarding my comment 3, the authors have confirmed that only one diploid strain (JP19) exhibits growth vigor. However, I did not see any response to my comment about why the diploid strains with elevated fitness as claimed by the authors are very scarce in the environment. Theoretically, if the intraspecific diploidization is adaptive, the diploid population should be expanded and it should be easy to find more JP19-like diploid strains in the environment. Is it possible that the diploid strain showing growth vigor in the in vitro tests performed in the lab actually does not have a competitive advantage in nature? I think this should be discussed in the Discussion section.

A minor comment: in the paragraph from L.281 to L.292 and the legend of Fig. 3C, "JP22" should be "R22".

Point-by-point responses to the reviewers' comments

Reviewer #3 (Remarks to the Author):

I appreciate the efforts of the authors to accommodate the majority of my concerns in the revised version. Regarding my comment 3, the authors have confirmed that only one diploid strain (JP19) exhibits growth vigor. However, I did not see any response to my comment about why the diploid strains with elevated fitness as claimed by the authors are very scarce in the environment. Theoretically, if the intraspecific diploidization is adaptive, the diploid population should be expanded and it should be easy to find more JP19-like diploid strains in the environment. Is it possible that the diploid strain showing growth vigor in the *in vitro* tests performed in the lab actually does not have a competitive advantage in nature? I think this should be discussed in the Discussion section.

Action. Apologies for oversight. As stated in the previous point-by-point response letter, we totally obtained six diploid isolates in this work. Through *in vitro* phenotypic assays, we confirmed only JP19 exhibited growth vigor. It is often assumed that the diploids via parasexual hybridization are often unstable, most of them could be returned to the haploid state through gradual loss of chromosomes. That is to say, the stable diploid hybrids in natural settings are very rare. Evolutionarily speaking, however, the adaptive diploid population should be expanded. Two possible explanations emerge. First, the limited sampling failed to identify more diploids. Second, the stable diploid hybrids may occur in unique habitats. We have discussed this point in the revised manuscript (lines 406-411).

We do believe that JP19 actually has a competitive advantage inferred from a wide range of abiotic stress assays. More importantly, when considering the effects of fungal inoculation on the plant growth under either normal or saline condition, we still found that JP19 had a pronounced plant growth promoting potential compared to its two parents (unpublished data). Thus, we think it is safe to conclude that JP19 displays obvious growth vigor, which could be further extended to confer more fitness to plant growth.

A minor comment: in the paragraph from L.281 to L.292 and the legend of Fig. 3C, “JP22” should be “R22”.

Action. Done.